

# Influence of climate variability, fire and phosphorus limitation on the vegetation structure and dynamics in the Amazon-Cerrado border

Emily Ane Dionizio da Silva[1], Marcos Heil Costa[1], Andrea Almeida Castanho[2], Gabrielle Ferreira Pires[1], Beatriz Schwantes Marimon[3], Ben Hur Marimon-Junior[3], Eddie Lenza[3], Fernando Martins Pimenta[1]

[1]Departamento de Engenharia Agrícola, Universidade Federal de Viçosa (UFV), Viçosa, MG, Brazil
[2]The Woods Hole Research Center, 149 Woods Hole Rd., Falmouth, MA 02540, USA
[3]Universidade do Estado de Mato Grosso, Campus de Nova Xavantina,
Nova Xavantina, MT, Brazil

*Correspondence to*: Emily Ane D. da Silva (emilyy.ane@gmail.com)

**Abstract**

Climate, fire and soil nutritional limitation are important elements that affect the vegetation dynamics in areas of forest-savanna transition. In this paper, we use the dynamic vegetation model INLAND to evaluate the influence of climate variability, fire and phosphorus limitation on the Amazon-Cerrado transitional vegetation structure and dynamics. We assess how each element affects the net primary production, leaf area index and biomass and compare the simulations of aboveground biomass to observed biomass map. We used two climate datasets - the 1960-1990 average seasonal climate and the 1948 to 2008 interannual climate variability, two regional datasets of total soil P content in soil, based on regional (field measurements) and global data and the INLAND fire module. Our results show that climate interannual variability, phosphorus limitation and fire occurrence gradually improve simulated vegetation types and these effects are not homogeneous along the latitudinal/longitudinal gradient showing a synergistic effect among them. In terms of magnitude, the effect of fire is stronger, and is the main driver of vegetation changes along the transition. The nutritional limitation, in turn, is stronger than



the effect of climate variability acting on the transitional ecosystems dynamics. Overall, INLAND
typically simulates more than 80% of the biomass variability in the transition zone. However, in many
places, the biomass is clearly not well simulated indicating that important soil and physiological factors
in the Amazon-Cerrado border, such as lithology and water table depth, carbon allocation strategies and
mortality rates, still need to be included in the model.



## 1 Introduction

The Amazon and Cerrado are the two largest and most important phytogeographical domains in South America. The Amazon forest has been globally recognized and distinguished not only for its exuberance in diversity and species richness, but also for playing an important role in the global climate by regulating water (Bonan, 2008; Pires and Costa, 2013) and heat fluxes (Shukla et al., 1990; Rocha et al., 2004; Roy et al., 2002). The Cerrado is recognized worldwide for being the richest savanna in the world (Myers et al., 2000; Klink and Machado, 2005). It is characterized by different physiognomies, ranging from sparse physiognomies to dense woodland formations, and the latter are commonly mixed with Amazon rainforest forming transitional areas. The Amazon-Cerrado transition extends for 6270 km from northeast to southwest in Brazil, and the ecotonal vegetation around this transition is a mix of the characteristics of the tropical forest and the savanna (Torello-Raventos et al., 2013).

Gradients of seasonal rainfall and water deficit, fire occurrence, herbivory and low fertility of the soil have been reported as the main factors that characterize the transition between forest and savanna globally (Lehmann et al., 2011; Hoffman et al., 2012; Murphy and Bowman, 2012). However, few studies have evaluated the individual and combined effects of these factors on Brazilian ecosystems ecotones (Marimon-Junior and Haridasan, 2005; Elias et al., 2013; Vourtilis et al., 2013).

It is challenging to assess the degree of interaction among these various environmental factors in the transitional region and to infer how each one influences the distribution of the regional vegetation. In this case, Dynamic Global Vegetation Models (DGVMs) can be powerful tools to isolate the influences of climate, fire and nutrients, therefore helping to understand their large-scale effects on vegetation (House et al., 2003; Favier et al., 2004; Hirota et al., 2010; Hoffman et al., 2012).



Previous modelling studies indicate that the Amazon rainforest could experience changes in
rainfall patterns which would either transform the forest into an ecosystem with more sparse vegetation –
similar to a savanna, what has been called as the "savannization of the Amazon" (Shukla et al., 1990; Cox
et al., 2000; Oyama and Nobre, 2003; Betts et al., 2004; Cox et al., 2004; Salazar et al., 2007) - or to a
seasonal forest (Malhi et al., 2009; Pereira et al., 2012; Pires and Costa, 2013). These studies had great
importance to the improvement of terrestrial biosphere modeling, but they neglect two important
processes in tropical ecosystem dynamics: fire occurrence and nutrient limitation.
In tropical ecosystems, fire plays an important ecological role and influences the productivity, the
biogeochemical cycles and the dynamics in the transitional biomes, not only by changing the phenology
and physiology of plants, but also by modifying the competition among trees and lower canopy plants
such as grasses, shrubs and lianas. Fire occurrence, depending on its frequency and intensity, may increase
the mortality of trees and transform an undisturbed forest into a disturbed and flammable one (House et
al., 2003; Hirota et al., 2010; Hoffmann et al., 2012). Fires also affect the dynamics of nutrients in the
savanna ecosystem, changing mainly the N:P relationship and phosphorus availability in the soil (Nardoto
et al. 2006).
Studies suggest that phosphorus is the main limiting nutrient within tropical forests (Malhi et al.,
2009; Mercado et al., 2011; Quesada et al., 2012) unlike the temperate forests. Phosphorus is a nutrient
that is easily adsorvided by soil due to the large amount of iron and aluminum oxides in the Amazon and
Cerrado acidic and strongly weathered soils (Dajoz, 2005; Goedert, 1986). In the tropics, the warm and
wet climate favors the high biological activity in the soil and the litter decomposition, not limiting the
nitrogen for plant fixation. In Cerrado, higher soil fertility is related to regions with greater woody plants



abundance and less grass cover, similarly to the features found in the Amazon rainforest (Moreno et al.,
2008; Vourtilis et al., 2013; Veenendaal et al., 2015). However, the phosphorus limitation is often
neglected by DGVMs that usually assumes unlimited phosphorus availability and consider nitrogen as
the main limiting nutrient. As it affects the dynamics between trees and grasses, the transition vegetation
between ecosystems may be misrepresented in such models.

In principle, in transitional forests, where the climate is intermediate between wet and seasonally

dry, the structure and phenology is heterogeneous among individuals which makes difficult its
representation. The Amazon-Cerrado border is the result of the expansion and contraction of the Cerrado
into the forest (see Marimon et al., 2006; Morandi et al., 2016), especially in the Mato Grosso state, where
extreme events, such as intense droughts, influence the vegetation dynamics (Marimon et al., 2014) and
the nutrient (Oliveira et al., *in press*) and carbon cycling (Valadão et al., 2016).

Currently, no model has demonstrated to be able to accurately simulate the vegetation transition

between Amazon and Cerrado. A better understanding of the main drivers that determine the distribution
of different vegetation physiognomies in the region is crucial for  more reliable simulations of the
transitional tropical ecosystems in future climate scenarios.

In this paper, we use the dynamic vegetation model INLAND (Integrated Model of Land Surface

Processes) to evaluate the influence of climate variability, fire occurrence and phosphorus limitation in
the Amazon-Cerrado transitional vegetation dynamics and structure. We assess how each element affects
the net primary production (NPP), leaf area index (LAI) and biomass and compare the simulations of
aboveground biomass (AGB) to an observed biomass map. The results presented here are important to



build models that accurately represent the actual transition vegetation, and show the need to include the
spatial variability of eco-physiological parameters in these areas.

## 2   Materials and methods

### 2.1   Study Area

The present study focuses on the Amazon-Cerrado transition (Figure 1). We use the official
delimitation of the Brazilian biomes proposed by IBGE (2004), and define five transects along the
transition border. Transects 1 to 4 were established considering approximately 330 km into the Amazon
and 330 km into the Cerrado domain, while Transect 5 is 880 km long on the southern Amazon-Cerrado
border. The transects are located as follows: Transect 1 (T1, 43°- 49°W; 5°- 7°S), Transect 2 (T2, 46°-
51° W; 7°-9S), Transect 3 (T3, 48°-54° W; 9°-11° S), Transect 4 (T4, 49° - 55° W; 11°-13° S), and
Transect 5 (T5, 53° - 61° W; 13°-15° S) (Figure 1).

### 2.2   Description of the INLAND Surface Model

The Integrated Model of Land Surface Processes (INLAND) is the land-surface component of the
Brazilian Earth System Model (BESM). INLAND is based on the IBIS model (Integrated Biosphere
Simulator, Foley et al., 1996) which considers changes in the composition and structure of vegetation in
response to the environment and incorporates important aspects of biosphere-atmosphere interactions.
The model simulates the exchanges of energy, water, carbon and momentum between soil-vegetation-
atmosphere. These processes are organized in a hierarchical framework and operate at different time steps,
ranging from 60 minutes to 1 year, coupling ecological, biophysical and physiological processes
(Kucharik et al., 2000). The vegetation structure is represented by two layers: upper and lower canopies,



and the composition is represented by 12 plant functional types (PFTs) (e.g., tropical broadleaf evergreen
trees or C4 grasses).

These PFTs can coexist within a grid cell and its tree annual LAI values indicate which vegetation

type dominates. To classify the vegetation type in a Tropical Evergreen Forest, the dominant PFT should
be a tropical broadleaf evergreen tree with annual mean leaf area index in upper canopy ($LAI_{upper}$) above
2.5 $m^2$ $m^{-2}$. To classify the vegetation type in a Tropical Deciduous Forest, the dominant PFT should be
a tropical broadleaf drought-deciduous tree with annual mean $LAI_{upper}$ above 2.5 $m^2$ $m^{-2}$. For ecosystems
where no tree PFT is dominant, the tree LAI between 0.8 and 2.5 $m^2$ $m^{-2}$ characterizes savannas, and
values smaller than 0.8 $m^2$ $m^{-2}$ characterize a grassland vegetation type.

We assume that the vegetation types Tropical Evergreen Forest and the Tropical Deciduous Forest

in INLAND represent the Amazon rainforest, while Savanna and Grasslands represent the Cerrado.
Savanna would be equivalent to the Cerrado physiognomies *Cerradão* and *Cerrado sensu strictu*, while
Grasslands would be equivalent to the physiognomies *Campo sujo* and *Campo Limpo* (*sensu* Ribeiro and
Walter, 2008).

The soil chemical properties are represented by the carbon cycle (C), nitrogen (N) and phosphorus

(P). The carbon cycle is simulated through vegetation, litter and soil organic matter, where the
biogeochemical module is similar to the CENTURY model (Parton et al., 1993; Verberne et al., 1990).
The amount of C existing in the first meter of soil is divided into different compartments characterized
by their residence time, which can vary in an interval of hours for microbial biomass and organic matter
to several years for lignin. For the nitrogen, the model considers only the soil N transformations and
carbon decomposition, not influencing the productivity of vegetation, i.e., there is a fixed C:N ratio.





Phosphorus is used only to limit the gross primary productivity. The total phosphorus available in the soil
($P_{total}$) is used to estimate the maximum capacity of carboxylation by the Rubisco enzyme ($V_{max}$) through
a linear relationship (Castanho et al., 2013).
$$V_{max} = 0.1013\, P_{total} + 30.037 \qquad\qquad (1)$$
where $V_{max}$ and $P_{total}$ are given in $\mu molCO_2\ m^{-2}\ s^{-1}$ and $mg\ kg^{-1}$, respectively.
INLAND also contains two fire model options for simulation. The first is a simple fixed-value
disturbance, which does not depend on the environmental conditions. The second is dynamic and based
on the fire module of the Canadian model of fire CTEM (Arora and Boer, 2005). In this module, all three
aspects of the fire triangle – the availability of fuel to burn, the flammability of vegetation depending on
environmental conditions, and the presence of an ignition source – are considered. It uses an arbitrary
anthropogenic fire probability which is added to the natural ignition probability. Only the dynamical fire
module is used in this study.
**2.3    Observed data**
**2.3.1    Phosphorus databases**
We used two phosphorus databases to estimate $V_{max}$ (Equation 1): one regional and one global
database (referred to as PR and PG, respectively). In addition, a control phosphorus map (PC) represents
the unlimited nutrient availability case, equivalent to a $V_{max}$ of 65 $\mu molCO_2\ m^{-2}\ s^{-1}$, or 350 mg P $kg^{-1}$ soil,
according to Equation 1.
The regional phosphorus database (PR) was developed from total phosphorus in the soil for the
Amazon basin published by Quesada et al. (2011) plus 54 additional available phosphorus samples (P




extracted via Mehlich-1 extractor) (Figure 2a). We used the $P_{-mehlich-1}$ and clay contents measured in a
forest-savanna transition region in Brazil (Mato Grosso state) to estimate $P_{total}$ and expand the coverage
area of the phosphorus data (Section S1). These 54 samples were gridded to a $1° × 1°$ grid to be compatible
with the spatial resolution used by INLAND, resulting in 12 new pixels with information of the total
phosphorus content (Figure 2a). For pixels without $P_{total}$ information, the PR dataset considered $P_{total}$ equal
to 350 mg P kg$^{-1}$ soil, similarly to the PC conditions.
A global dataset of total phosphorus content in the soil ($P_{total}$) was also used (Figure 2b). This
global total phosphorus data is part of a database containing six global maps of the different forms of
phosphorus in the soil (Yang et al., 2013). The total phosphorus was estimated from lithologic maps,
distribution of soil development stages, fraction of the remaining source material for different stages of
weathering using chronosequence studies (29 studies), and phosphorus distribution in different forms for
each soil type based on the analysis of Hedley fractionation (Yang and Post, 2011), which are part of a
worldwide collection of soil profile data. According to Yang et al. (2013), the uncertainties and limitations
of this database are restricted to the Hedley fractionation data used. When quantified, these uncertainties
are 17% for low weathered soils, 65% for intermediate soils and 68% for highly weathered soils.
**2.3.2    Above-Ground Biomass (AGB) database**
The AGB database used was created by Nogueira et al. (2015) and considers a vegetation
originally existing in the Brazilian Amazonia. This database was based on a vegetation map at a scale of
1:250000 (IBGE, 1992) and biomass averages from 41 published studies that had conducted direct
sampling in either forest (2317 plots) or non-forest or contact zones (1830 plots). We bilinearly



interpolated the AGB (dry weight) for each transect considering $1° \times 1°$ to ensure compatibility of the
observed and simulated data.

The five-fixed longitudinal transects (Figure 1) were used separately to characterize vegetation

from AGB in the Amazon-Cerrado border (Figures 3a and 3b). These transects were also compared to the
simulated results considering the different combinations among all treatments.

In general, the higher AGB values in the west and lower values in the east for T1, T2, T3 and T4

are consistent with the transition from a dense and woody vegetation towards a sparse vegetation with
lower biomass, typical of the Amazon-Cerrado border (Figure 3a). However, in T1, a gradual reduction
of biomass along the west to east gradient is shown, differently than T2, T3 and T4 where the biomass
decreases abruptly. In T5 no west-east gradient is present with high biomass heterogeneity and
predominant low biomass across the transect (Figure 3b).

In the north of the Amazon-Cerrado border (T1), the duration of the dry season is smaller and the

annual rainfall is higher in comparison to other transects. This region is characterized by the presence of
transitional forest formations, such as dense forest and a transitional deciduous/semideciduous forest.
These forest formations are common in the south of Tocantins state, where water is not limiting, the soils
are sandy and the surface is flat (Haidar et al., 2013). Thus, the less pronounced biomass decline could
feature the transition from two forest formations with different structures. On the other hand, T5 located
at the south of the transition border, is exposed to longer dry season duration and stronger precipitation
seasonality. Despite its lower biomass. T5 presents a high AGB spatial variability (Figure 3b). Due to its
greater territorial extent (880 km) in the Cerrado domain, different types of vegetation have been sampled,
featuring a mixed medium-sized and small vegetation. Moreover, different soil and topographical settings





are found, featuring a highly heterogeneous landscape in this region (Silva et al., 2006). For example, in
the Cerrado domain, the vegetation structure varies substantially within and between vegetation types,
and generally has been associated with soil spatial patterns, landform and drainage (Silva et al. 2006).

### 2.4 Simulations

The model was forced with prescribed climate based on the Climate Research Unit (CRU)
database (Harris et al., 2014). Two climate boundary conditions were used: the first is referred to as the
monthly climatological average (CA) that represents the average climate of 1961-1990. The second
climate boundary condition is the historical dataset, for the continuous period between 1948 and 2008
(CV). For both boundary conditions, the variables used are rainfall, solar radiation, wind velocity and
maximum and minimum temperatures. The dataset has a 1-degree spatial resolution and a monthly time
resolution.
Soil texture data is based on the IGBP-DIS global soil (Global Soil Data Task 2000) (Hansen and
Reed, 2000). The model simulations were run for a total period of 427 years. An initial spin up of 366
years was used to initialize the carbon pools to equilibrium (equivalent to the period 1582-1947), when
the climate data for 1948-2008 was applied cyclically, plus the period of 1948-2008.
The experiment design was based on the combination of climate scenarios (CA, climatological
average, 1961-1990; CV, monthly climate data, 1948-2008), variable atmospheric $CO_2$ concentration
(from 278 to 380 ppm), the nutrient limitation on $V_{max}$ (PC, no phosphorus limitation
($V_{max} = 65 \, \mu molCO_2 \, m^{-2} \, s^{-1}$); PR, regional phosphorus limitation; PG, global phosphorus limitation) and
the occurrence of fire effects (F) or not (Table 1).



These combinations allow the evaluation of individual and combined effects of climate, soil
chemistry, and the incidence of fire in the total productivity of the ecosystems, tree biomass and LAI, and
their effects on the location of the Amazon-Cerrado border.
We simulated the individual effects of climate variability, phosphorus limitation and fire on the
variables: Net Primary Production (NPP), tree biomass, and LAI of the upper and lower canopies
($LAI_{upper}$, $LAI_{lower}$).
The analysis of the isolated effects of climate variability was performed using a combination of
CV+PC and CA+PC. We consider that the subtraction between the simulations (CV+PC)-(CA+PC) =
$(CV-CA)|_{PC}$ represents the effect of climate variability if phosphorus limitations are not considered.
The same logic was applied to isolate other factors such as fire and phosphorus in different climate
scenarios. For example, the isolated effect of fire with an average climate scenario without the influence
of phosphorus limitation is calculated by the difference between CA+PC+F and CA+PC, so that
(CA+PC+F) - (CA+PC) = $F|_{CA, PC}$. The isolated effect of fire with a climate variability scenario without
influence of P limitation is calculated by the difference between CV+PC+F and CV+PC, so that
(CV+PC+F) - (CV+PC) = $F|_{CV, PC}$.The different combinations of climate scenarios with or without the
fire effect and different phosphorus limitations are described in Table 2.
**2.5    Statistical analysis and determination of the best model configuration**
The variables tested are NPP, LAI and AGB. The statistical analysis is divided in four parts. First,
we present maps with the spatial patterns of isolated effects for all simulated area, and the differences
between the treatments that are statistically significant at $p<0.05$ (according to the t-test) are shaded.





Second, we present an analysis of variance using the one-way ANOVA and the Tukey-Kramer
test in the transition zone. To this end, we grouped treatments according to climate (Group 1), nutrient
limitation (Group 2) and presence or absence of fire (Group 3). In Group 1 we tested if CA and CV results
were significantly different from each other along the transects, regardless of nutritient limitation,
presence or absence of fire. Similarly, in Group 2 we organize the simulations according to the type of
nutritient limitation, PC, PR or PG, regardless of climate, and presence or absence of fire. In Group 3 the
simulations were grouped considering fire presence or no fire. Finally, all treatments were tested
individually to assess the effects on NPP, LAI and AGB.
Third, a correlation coefficient between the simulated and observed values for AGB was
calculated for each transect. The simulated variables are averaged for the last 10 years of simulations
(1999 - 2008) and compared to biomass from Nogueira et al. (2015) within a grid cell.
Finally, we evaluate INLAND's ability to assign the type of dominant vegetation analyzing 10
years of probability of occurrence. If the dominant vegetation type (evergreen tropical forest or deciduous
forest for the Amazon rainforest, and savanna or grasslands for Cerrado) in a pixel is the same in more
than 90% of the simulated years (9 of 10), then the simulated vegetation type is defined as "very robust"
for that pixel; if it occurs in 70 - 90% of the simulated years, it is considered to be "robust". If the dominant
vegetation occurred in less than 70% of simulated years, the pixel is considered "transitional" vegetation.



## 3 Results

### 3.1 Influence of climate, fire and phosphorus in the Amazon-Cerrado transition region

#### 3.1.1 Spatial patterns

Overall, the inclusion of climate interannual variability (CV) resulted in a decrease in the simulated average biomass by 3.8% in Amazonia, and by 8.7% in Cerrado in comparison to average climate (CA) (Figure 4a). The differences between CV and CA AGB simulations are statistically significant and range from -3 kg-C m$^{-2}$ to 2 kg-C m$^{-2}$. The state of Pará, where geographically there is higher influence of El Niño phenomenon, presented the highest decrease in AGB in CV simulation. In the state of Roraima, on the other hand, there was an increase of about 2 kg-C m$^{-2}$ in AGB when CV was considered. Bolivia and southwest of Mato Grosso state also presented, in some grids points, a significant increase in AGB higher than 2 kg-C m$^{-2}$.

The phosphorus total concentration acts in average as a limiting factor in the tree biomass. Biomass decreased by 13% in regional phosphorus database (PR) and 15% in global phosphorus database (PG). In PR, biomass decreased mainly in the southeastern Amazonia (between Pará and northeastern of Mato Grosso states) and northwestern Amazonas state (Figure 4b). In PG, the largest declines in biomass occurred in central Amazonia, northeastern Pará and northeastern Mato Grosso (Figure 4c). In Cerrado, on the other hand, tree biomass declined by 2% and 9% in relation to control simulation when PR and PG were considered, respectively. In PR the few pixels in the Cerrado that have nutrient limitation showed a significant decrease in tree biomass (Figure 4b), but in PG on the other hand, our results showed biomass reduction statistically significant for most of the Cerrado domain, except only for the southern Tocantins



state (Figure 4c), reinforcing the hypothesis that phosphorus limitation influences the vegetation structure

and landscape, and that we need to implement P cycles in tropical ecosystems modeling.

Fire effect declines tree biomass with largest magnitude and intensity than phosphorus limitation

or interannual climate variability (Figure 4d), as expected. The small or null fire effect in the Central

Amazon rainforest shows that the Amazonia forest is naturally inflammable as well as a gradient towards

seasonally dryer climate that increases the intensity and magnitude of fire effects towards the Cerrado

(Figure 4d). The fire effect over the Amazon domain was 8-9% of the P limitation effect (PR and PG,

respectively), while the fire effect over the Cerrado was over 250% of the P effects. This magnitude

characterizing the quickly increase of grasses after fire occurrence in the latter.

### 3.1.2 Influence of climate, fire and phosphorus in the transects

Results of the ANOVAs and Tukey–Kramer test indicate that the inclusion of climate interannual

variability (CV), limitation by phosphorus (PR and PG) and fire in INLAND model led to significantly

different averages of NPP, LAI and biomass in transition zone. This influence of climate, phosphorus and

fire are shown separately in Tables 3 to Table 5 and combined in Table 6.

The effects of climate and phosphorus on productivity show that CV reduce the NPP from 0.68

kg-C m$^{-2}$ yr$^{-1}$ to 0.64 kg-C m$^{-2}$ yr$^{-1}$ (Table 3) and the phosphorus effect (PR and PG) result in biomass

decline from 0.71 kg-C m$^{-2}$ yr$^{-1}$ and 0.64 kg-C m$^{-2}$ yr$^{-1}$, respectively (Table 4). The fire effect, moreover,

has a positive effect increasing the NPP from 0.66 kg-C m$^{-2}$ yr$^{-1}$ and 0.67 kg-C m$^{-2}$ yr$^{-1}$ for PR and PG,

that although it is low, is statistically significant according to Tukey Kramer test (Table 5).

The same relationship was found in LAI$_{total}$ and LAI$_{upper}$ where CV and nutrient limitation reduce

the LAI$_{total}$ in the canopy (Table 3 and Table 4). For LAI$_{total}$ the effect of fire is positive, increasing tree



times LAI$_{lower}$ and increasing LAI$_{upper}$ (Table 5). In biomass, the magnitude of fire effect is greater in
relation to the climate variability and nutritional limitation effects. While climate variability introduced
changes in the order of 5% for AGB (Table 3) and phosphorus changes of up to 14% (Table 4), fire
decreased AGB by 46.7% (Table 5).

The results of NPP and AGB for CV+PC simulation, do not present significant difference from

CA+PC (Table 6) which does not agree with Table 3. LAI$_{total}$ and LAI$_{upper}$ have significant reduction for
transitional area when interannual climate variability was considered, which could be improve the
INLAND vegetation classification results.

The inclusion of P limitation in INLAND simulations (CV+PR and CV+PG) without fire

occurrence caused significant decline in relation CV+PC for all variables (NPP, AGB, and LAI$_{total}$), but
no difference between them (PR, PG) (Table 6).

Fire effects are significant only for structural variables as AGB, LAI$_{total}$, LAI$_{upper}$ and LAI$_{lower}$. It

presents an increase of LAI $_{total}$ of 1.52 m$^2$ m$^{-2}$ in CV+PG+F in relation to CV+PG, and of 1.32 m$^2$ m$^{-2}$ in
CV+PR+F in relation to CV+PR (Table 6). This increment of LAI corresponds to the increase of grasses
and the decrease of trees.

### 3.1.3    West-East patterns of AGB in the Amazon-Cerrado transition

The model used in this study simulates > 80% of the observed biomass variability in all treatments

along the transition area except in T5 (Table 7). It shows that the model is able to capture biomass
variability along the transition area, which is relevant when compared to recent studies that validate the
representation of models in the Amazon using biomass observation data (Senna et al., 2009; Castanho et
al., 2013).



It is not possible to identify a treatment that best represents the biomass of all transects (Table 7).
A combined analysis of Table 7 and Figure 5 indicates a general agreement that biomass decreases from
W to E in T1 to T4, and this is well captured by several configurations of the model, with specific
differences among them. Overall, CA an PC configurations yield higher biomass, while the introduction
of interannual climate variability (CV), phosphorus limitations (PG) and fire (F) reduces the biomass.
However, the simulated results may be above or below the observed ones, which suggests an additional
local factor not included in the model.
Most likely, errors may have occurred in the representation of soil physical properties, which are
very important in the calculation of moisture stress.
The curves of AGB (Figure 5) shows the impact of CV, phosphorus limitation and fire along the
West-East transition. The AGB gradient of CA+PC and CV+PC are similar in all transects showing that
in the transition the climate interannual variability into the model has little influence on vegetation
structure. On the other hand, the phosphorus limitation presented high influence in the transition,
decreasing the ABG especially in the grids in the west transects, where the Amazon vegetation is
predominant. This influence could be easily observed in T3 and T4, where PG decrease the AGB by
2 kg-C m$^{-2}$ in the west of the transects (Figure 5). In T1, T2 and T5 AGB decline is also higher when P
was incorporated in relation the curves limited only by CV. For these transects INLAND represents the
gradient of dense vegetation structure towards a sparse vegetation decreasing the AGB along the transects.
In T1, model simulations tend to underestimate the highest and the lowest biomass extremes, despite of
the improvement in the correlation with the inclusion of the fire component the absolute values were all
underestimated in this transect.



Fire, however in T2, T3 and T4, is responsible to approach the simulated biomass to the observed
biomass in the east grids into Cerrado domain. In T5, these relations are similar, with climate presenting
less influence on biomass decrease than phosphorus, and fire appears mainly as a reducer factor.
**3.2    Simulated composition of vegetation**
Most of the pixels in CA show very robust simulations, with more than 90% of the same vegetation
cover in the last 10 years simulated (Figure 6a-c and 6g-i). A larger number of pixels with values below
70% of agreement of vegetation type were simulated when considering CV. An even higher variability in
CV compared to CA simulations was observed when we added the effects of phosphorus limitation and
fire.
The vegetation composition in all nutrient limitation scenarios for CA simulations resulted in
>90% of agreement on the vegetation type for nearly all pixels, except for the north of Cerrado domain
(Figures 6a, 6b and 6c). CA+PC and CA+PR simulations had the same vegetation composition, while
CA+PG replaced the deciduous forest by evergreen forest in the central Cerrado region, around 8° S 46°
W (Figures 6A, 6B and 6C). Cerrado was better represented in CV+PC, CV+PR and CV+PG than in the
same CA combinations (Figure 6). The occurrence of forested areas in the central Cerrado decreased in
CV combinations, these being replaced by the savanna or grassland vegetation class.
When the effect of fire was added to CA simulations, the model simulated an increase in the
uncertainty on the vegetation cover classification in the Cerrado region. The effect of fire reduced the
presence of deciduous forest in central Cerrado biome as well as in CA+PC, and the vegetation was
replaced by evergreen forest and savanna in CA+PC+F (Figures 6G, 6H, 6I). In CV simulations, fire



effect results in the replacement of the deciduous and perennial forest by savanna and grasses in all central
Cerrado region (Figures 6J, 6K and 6L).

Transitional forest areas in the northern Tocantins state (TO) and southeastern Mato Grosso (MT)

are not adequately represented. With >90% of concordance, INLAND assigns the existence of tropical
evergreen forest rather than deciduous forest in TO for all combinations used, and the existence of tropical
evergreen forest rather than savanna in MT, indicating difficulty in simulating transitional vegetation in
this place.

**4    Discussion**

The inclusion of climate interannual variability (CV), limitation by phosphorus (PR and PG) and

fire in INLAND model led to significantly different averages of tree biomass (Figure 4) and aboveground
biomass (Table 6), which agrees with the consensus that these factors influence vegetation structure in
the forest-savanna border (Hoffmann et al., 2012; Dantas et al., 2013; Lehmann et al., 2014; Baudena et
al., 2015). In this work, the spatial analysis and the Tukey-Kramer test (TK) results show that there is a
difference in magnitude among these factors in vegetation, with fire occurrence and phosphorus limitation
being stronger than climate interannual variability (Figure 4).

CV declines biomass predominantly in eastern Amazonia (Figure 4a). In this region, the El Niño–

Southern Oscillation (ENSO) phenomenon could reduce by 50% the amount of rainfall, putting the
vegetation under intense water stress (Botta and Foley, 2002). This change in interannual rainfall brings
in direct changes in carbon flux and stocks in leaves and wood leading changes in vegetation structure.

The differences between CA and CV were significant for most part of these biomes, except central

Amazonia (Figure 4a), where the climate interannual variability and seasonality of precipitation have





been pointed as secondary effect on vegetation, since there is no shortage of water availability during the
dry season (Restrepo-Coupe et al., 2013). However, although CV has a significant effect on AGB in our
region-wide results (Figure 4a), there is no significant difference along the transects in the Amazon-
Cerrado transition (Table 6). This difference occurred because a smaller sample size was used to test CV
in each simulation. In Table 6, the difference between CA+PC and CV+PC is around 0.30 kg-C m$^{-2}$, not
significant at $\alpha=0.05$. On the other hand, when we analyzed the influence of CV in the same place, but
using all simulations, regardless of phosphorus limitation and fire occurrence, the results showed a
significant decrease in AGB by 0.38 kg-C m$^{-2}$ in the transition (Table 3). Thus, we conclude that CV has
influence on the vegetation structure, but its magnitude is so small along the transition that it is not
detected by Tukey-Kramer test in the CV+PC simulation. Thus, compared to P limitation and fire
occurrence, CV has a secondary role. It was expected, since a seasonal climate region, such as the
Amazon-Cerrado transition, suffers less influence of interannual oscillations in precipitation and the
vegetation is likely more adapted to the water stress.

Along the Cerrado, lower water availability in CV affects tree biomass although that vegetation is

predominantly grassy-herbaceous. The biomass decline is significant for most part of the simulated
Cerrado domain (Figure 4a) and average values could reduce by half the amount of tree biomass in
Amazonia. In this biome, the shallow soils, high porosity and the fine root biomass exposed the few trees
to higher water stress compared to the Amazon in dry years (Ruggeiro et al., 2002).

Phosphorus limitation effect was statistically significant for PR and PG along all the Amazon

domain and the main differences between these simulations were the spatial patterns of tree biomass
decrease (Figure 4b and Figure 4c). We cannot affirm which of these databases are better because they





are the results of different methodologies and observations (Yang et al., 2014; Quesada et al., 2009).
However, we observe that PG showed a higher biomass decrease in central Amazonia, northeastern Pará
and northeastern Mato Grosso, indicating that in these areas the phosphorus limitation is higher. This
result does not corroborate the northwest-southeast AGB gradient found in the Amazon basin, which is a
result of higher productivity in the west where soils are more fertile than those found in the southeast
(Aragão et al., 2009; Saatchi et al., 2007; Nunes et al., 2012; Lee et al., 2013). On the other hand, PR
biomass agrees with the northwest-southeast gradient, presenting less limitation in the soils of central
Amazonia with declines in biomass mainly in the southeastern Amazonia (between Pará and northeastern
of Mato Grosso states) (Figure 4b).
In Cerrado, phosphorus limitation also limited vegetation when was considered in model
simulations (Figure 4c) and presented statistically significant differences in relation to CV+PC. In this
biome, as well as in the Amazon, tree abundance has been generally associated with increases of soil
fertility (Vourtilis et al., 2013). Richness and diversity of vegetation was correlated with soil fertility
(Long et al., 2012), and highlight the importance of P in the composition and maintenance of vegetation,
especially in transition areas.
Compared to the Amazon domain, the effects of phosphorus limitation is lower in the Cerrado,
probably due to the adaptation of savanna species to the P-restrictions caused by a lack of an effective
and robust nutrient cycling system compared to the forests (Oliveira et al. *in press*). In PR, the few pixels
that have nutrient limitation showed a significant decrease in arboreal biomass (Figure 4b), but in PG, our
results showed reduction of biomass statistically significant for most of the Cerrado domain, except only
for the southern Tocantins state (Figure 4c). Despite the differences in spatial patterns, there was no





statistically significant differences between PR and PG within the transects (Table 4) showing that there
is little difference in west-east gradient of soil fertility and that INLAND has small sensitivity to the
different P databases used.

The spatial difference between the global phosphorus database (PG) and the regional phosphorus

database (PR) showed that PG underestimates total phosphorus in the western Amazonia, and
overestimates in northern Amazonia with respect to PR. Moreover, PG presented underestimates in south
of the transition and overestimates in Cerrado domain and northeast of transition at transects limits when
compared to PR (Figure S1). These differences show that the understanding of phosphorus influence
limitation, on vegetation is directly associated to reliability of databases and reveal the need to improve
the data set for $P_{total}$ in the soils of the Amazon and Cerrado/Amazon transition domains.

The $P_{total}$ versus $V_{max}$ linear relationship is a proxy of the influence of soil fertility in

photosynthesis, not considering the phosphorus-soil-vegetation feedback. To our knowledge, none of the
current Dynamic Global Vegetation Models DGVMs considers the complete cycle of phosphorus, despite
the importance of nutrient cycling for the biomass maintenance and dynamics of the tropical vegetation
in dystrophic soils. For example, nutrient cycling in the Amazon/Cerrado transition is close related to the
hyperdynamic turnover of the biomass (Valadão et al. 2016), in which some key species might also be
key to the hypercycling of nutrients through which vegetation sustain the constant input of nutrients,
including large annual amounts of available P (Oliveira et al. *in press*).

The decrease in tree biomass occasioned by phosphorus limitation can contributes to a decrease

in litter production and consequently could affects the nutrient cycling in tropical ecosystems. According
to Oliveira et al. (*in press*), the litter produced by vegetation corresponds to the main return route to the



available fraction of phosphorus for plants, especially in the transition areas, where $P_{available}$ in the soil is
very low. In our model, however, phosphorus acts directly in the photosynthesis limitation through $V_{max}$
and cannot be reabsorbed by the roots. Thus, the litter produced in vegetation contribute only to dry matter
and fire occurrence increase. The litter affected by fire occurrence volatilizes the small amount of
phosphorus available to plants, increasing the nutrient losses of the ecosystem. Despite this simplified
representation in INLAND, it is clear that phosphorus has a significant effect on woody biomass in the
Amazon and Cerrado and that biomass spatial variability is fully associated with the P variability in the
soil (Figure 4b and Figure 4c).

The comparison between (CV+PG)–(CV+PG+F) showed that Amazon vegetation remains

naturally impervious to fire (Figure 4d), contrary to the tropical savannas, which are naturally affected by
this disturbance, followed by natural recover (i.e. resilience)  (Lehmann et al., 2011; Murphy and
Bowman, 2012;  Lehmann et al., 2014).  The fire occurrence along the Amazon-Cerrado border is
influenced by the gradient from a dense forest towards a seasonal and/or semi deciduous forest, or a sparse
vegetation type like savanna.

Our results confirm our hypothesis showing fire as an important factor controlling the biomass

dynamic (Silvério et al., 2013; Couto-Santos et al., 2014; Balch et al., 2015), with statistically significant
influences (Table 5). The fire effect may reduce AGB by 50% in the transition zone, which under climate
change or deforestation conditions, may lead to an even stronger change in the vegetation structure and
dynamic.

In the Cerrado domain, the fire effect implies in significant increases of shrubs and herbaceous

vegetation and decreases of the arboreal component. The Cerrado, however, is relatively resilient to fire



depending on the velocity, intensity and duration of the burning (Rezende et al., 2005; Elias et al., 2013
Reis et al., 2015). The adaptive morphological nature and the low nutrient requirement of vegetation allow
Cerrado the ability to rapidly restore the vegetation after fire occurrence (Hoffmann et al., 2005;
Hoffmann et al., 2012).

In the transect zone, our model was able to represent the variability of the biomass. Compared to

previous modeling studies, our study shows an improvement in the correlations between simulated and
observed AGB. Senna et al. 2009 found 0.20 as maximum correlation coefficient between simulated and
observed ABG while Castanho et al. (2013) showed 0.80 for Amazonia domain. In this study, the
correlation coefficients for ABG are usually above 0.80 for the transects, regardless of the treatment,
except for T5 (Table 7).

It is clear that interannual climate variability, fire occurrence and phosphorus limitation in the

transition zone reduce the biomass and play significant roles in the simulations (Figure 5), but the only
inclusion of these effects is still insufficient to represent the actual vegetation structure in the Amazon-
Cerrado border. In our interpretation, this means that other important factors are still missing from the
simulation, and that our P dataset and fire module need to be improved. A better spatial representation of
soil physical properties, including shallow rocky soils, as well as physiological characteristics of the
vegetation as carbon allocation, deciduousness of vegetation, and residence time are probably needed to
improve this simulation.

In the transects, the influence of climate interannual variability (CV+PC), P limitation (CV+PG)

and fire (CV+PG+F) on AGB are also evident, when compared to the control simulation (CA+PC) and
to the observed data (Figure 5). For all transects the biomass curves have similar patterns, where the



smaller difference is observed between CA+PC and CV+PG curves, while the larger difference is when
fire is incorporated in the Amazon. The effect of phosphorus limitation appears as an intermediate effect
reducing the biomass more than the effect of climate variability. In Cerrado, it is observed that there is
little or no difference among biomass simulated by CA+PC, CV+PC and CV+PG, revealing that in this
biome, interannual climate variability and phosphorus have smaller influence in the vegetation structure.
However, in the east of T2, T3 and T4, fire is the factor that adjusts the simulated to the observed data
(Figure 5), differently than the grid points in the west, where CV+PG is a better proxy between observed
and simulated data.

Such conditions are interesting because reflects the different mechanisms that regulates these

ecosystems and probably the phytophysiognomies distribution. For example, P limitation seems to be the
factor that improves biomass simulated in regions where the predominant vegetation type is the tropical
rainforest. Fire, on the other hand, improves the biomass in grid points where the Cerrado occurs.
Moreover, important factors such as productivity partitioning into leaves, roots and wood carbon pools
are assumed to be fixed in space and time within a given PFT, neglecting the natural capacity of
transitional forests to adapt itself and to adjust their metabolism to seasonal climate conditions (Senna et
al., 2009). In years of severe drought, transitional forests could prioritize the stock of carbon to fine roots
instead of the basal increment in order to maximize access to available water, or make hydraulic
redistribution to maintain the greenness and photosynthesis rates (Figure 5a). Brando et al. (2008) found
high sensitivity in carbon allocation for eastern Amazon basin trees, which reduced wood production by
13-60% in response to an artificial drought. Although in INLAND soil moisture is able to reduce the



photosynthetic rates during the months of lower rainfall, it does not dynamically change the allocation
rates, exposing the PFTs to severe water stress and underestimating the biomass (Figure 5a).
Along this seasonal climate, located in the central region of the Amazon-Cerrado transition, T3
and T4 showed the highest average correlations between observed and simulated data (Table 7). For these
transects, INLAND is able to capture the high variability of biomass gradient. However, the AGB in the
Cerrado grid points of these transects are underestimated due to the lower water availability in this
environment (Figure 5b and Figure5d).
At the T5 transect located at the south of the transition, the average correlations were low for all
treatments, indicating that INLAND has difficulty to represent the AGB gradient there (Table 7).
However, it captures the lower biomass as compared to the northern ones. In this region, the vegetation
is characterized by a wide diversity of physiognomies, which varies with other preponderant factors, such
as lithology, soil depth, topography and fertility. The observed data also showed high biomass variability,
indicating that there are changes in the vegetation structure, featuring medium-sized and small vegetation
types on different soil types. In INLAND, however, features such as lithology and water-table depth are
not considered due to the complexity of its representation on the large scale, limiting the representation
of a heterogeneous environment throughout the transition. Thus, we observe for T5 transect an
underestimated biomass and a high variability along the west-east gradient.
Overall, we observe that different patterns of vegetation distribution along the Amazon-Cerrado
border exist and are influenced not only by climate variability, phosphorus limitation, and fire, but also
by ecophysiological parameters, which have different behavior according to the environmental conditions
and soil proprieties. Obtaining these parameters is a challenge to the scientific community once the field



measurements are difficult due to the extension of the transition area. More observed data are needed to
establish and implement the plasticity of the fixed parameters such as carbon coefficients allocation,
residence time, dependence of the deciduousness to phosphorus, and others, to improve the representation
of the Amazon-Cerrado limits, since the model showed that it is able to represent the average behavior of
the transition.

The simulation of the spatial distribution of vegetation types by INLAND also improved when

climate, phosphorus limitation and fire were considered. The simulations with CV showed a decrease in
evergreen forests areas into Cerrado domain with replacement by tropical deciduous forest and Savanna
(Figure 6). The inclusion of fire in CV simulations brings an improvement along the Amazon-Cerrado
border, enlarging the savanna vegetation type and decreasing forests areas into Cerrado border domain.
The best combination found to represent the frontier of the Amazon-Cerrado was CV+PG+F (climate
variability + global map of phosphorus limitation + fire occurrence). Although CV+PC+F and CV+PR+F
presented similar vegetation types to CV+PG+F, the latter showed more robust results in the Cerrado
domain. On the other hand, in the absence of fire, the main factor driving the distribution of vegetation in
terms of biomass and consequently of phytophysiognomic types, becomes the availability of P in the soil.
Therefore, the fire dynamic is the preponderant factor controlling the dynamics and shifts of the
vegetation in the Amazon/Cerrado transition zone
**5    Conclusions**

This is the first study that assesses the modelling representation of the Amazon-Cerrado border.

This study shows that, although INLAND forced by a climatological database is able to simulate basic
characteristics of the Amazon-Cerrado transition and that adding factors such as climate interannual





variability, phosphorus limitation and fire gradually improves simulated vegetation types, improvements
are still needed. These effects are not homogeneous along the latitudinal/longitudinal gradient, which
makes the adequate simulation of biomass challenging in some places along the transition. Our work
shows that there is a synergistic effect between these variables in Amazonia-Cerrado transition and that,
in terms of magnitude, the effect of fire is stronger than the nutritient restriction and climate, resulting in
the main determinant factor of the vegetation changes along the transition. The nutrient limitation in turn
is stronger than the effect of climate variability acting on the dynamics of transitional ecosystems,
therefore determining the vegetation distribution in the absence of fire.

Overall, although INLAND typically simulates more than 80% of the variability of biomass in the

transition zone, in many places the biomass is clearly not well simulated. Situations for wet or dry
conditions were well simulated, but the simulations are generally poor for transitional areas where the
environment selected physiognomies that have an intermediate behavior, as is the case of the transitional
forests in northern Tocantins and Mato Grosso. The lack of field parameters measured in the transition
zone is still a major limitation to improve the DGVMs. Spatially explicit carbon allocation strategies,
mortality rates, physiological and structural parameters are necessary to establish numerical relationships
between the environment and the vegetation dynamic models to make them able to correctly simulate
current patterns and future changes in vegetation considering future climate change. In addition, it is also
needed to include not only the spatial variability, but also temporal variability in physiological parameters
of vegetation, allowing a more realistic simulation of the vegetation-climate relationship. Finally, our
results reinforce the importance and need of the DGVMs to incorporate the nutrient limitation and fire
occurrence to obtain more realistic results.





## 6 Acknowledgements

We gratefully thank FAPEMIG and CAPES for their financial support.

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





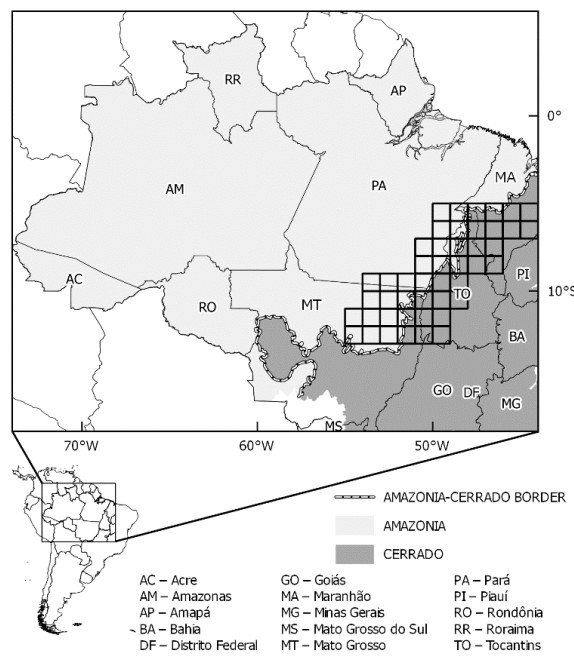

**Figure 1.** Delimitation of the study area Amazonia (in light gray) and Cerrado (in dark gray) (IBGE

(2004), and the location of five transects used in this work (from T1 to T5). The dashed line represents

the border between biomes.





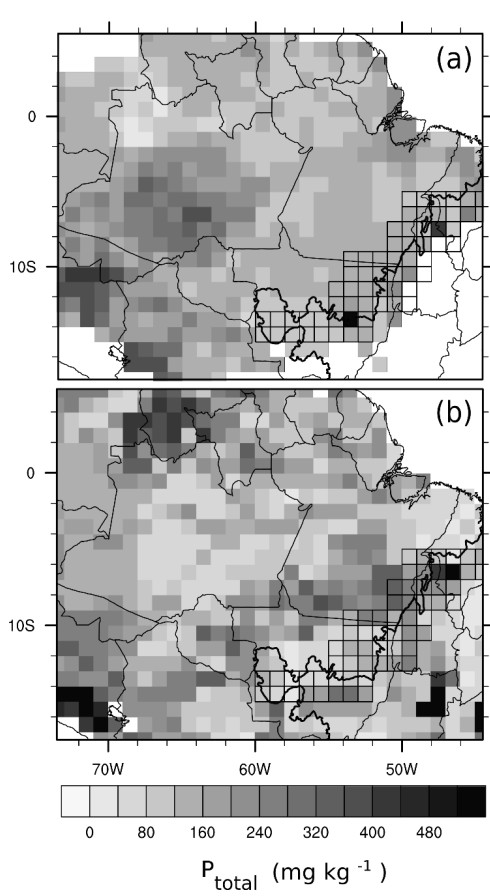


**Figure 2.** (a) Map of regional total phosphorus in the soil with new estimated $P_{total}$ data - PR, (b)
Map of global total phosphorus in the soil (Yang et al., 2013) – (PG).





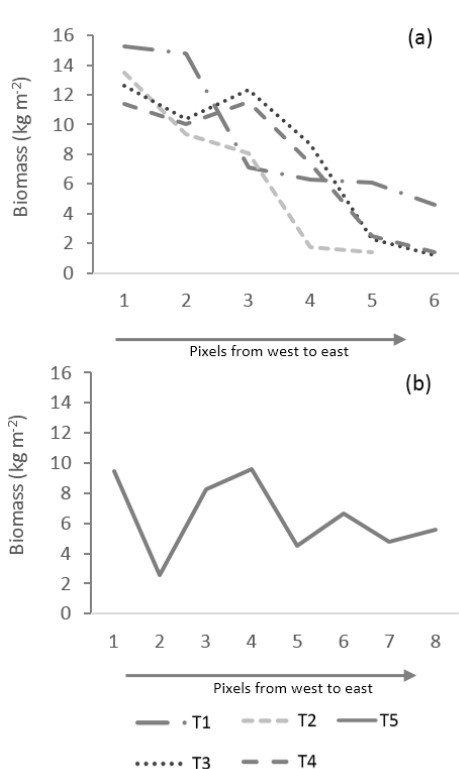


**Figure 3**. West-East patterns of AGB in the Amazonia-Cerrado transition for the five transects

analyzed.





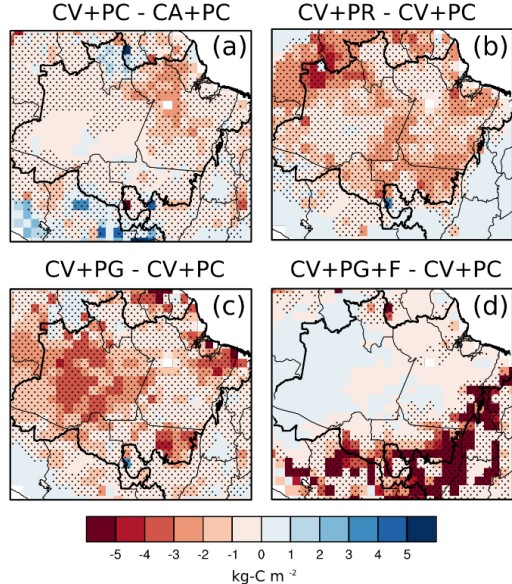


**Figure 4.** Interannual climate variability effect (a), Regional phosphorus limitation effect (b), Global

phosphorus limitation effect (c), and fire effect (d) on tree biomass. The hatched areas indicate that the

variables are significantly different compared to the control simulation at the level of 95% according to

the t-test and the thick black line is the geographical limits of the biomes.



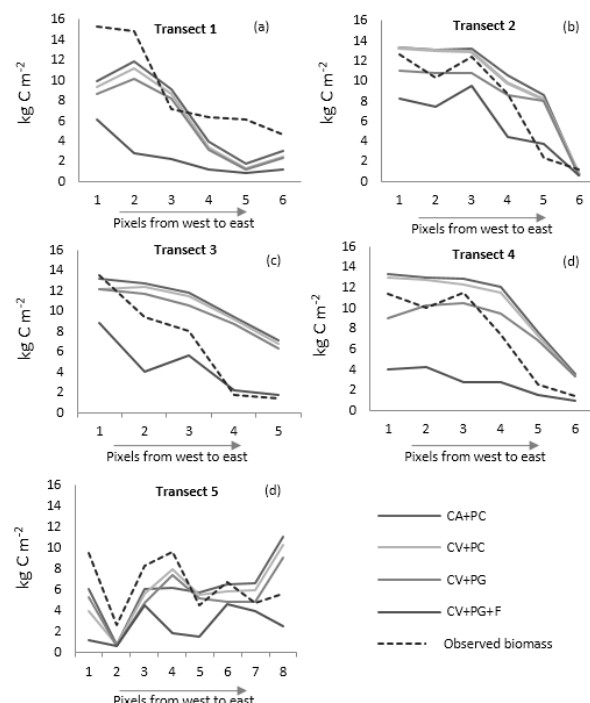

836

**Figure 5.** Longitudinal biomass gradient in Amazonia-Cerrado transition simulated for T1 to T5

considering different combinations: observed data; seasonal climate control simulation (CA+PC);

interannual climate variability (CV+PC); interannual climate variability + global phosphorus limitation

(CV+PG); and interannual climate variability + phosphorus + fire occurrence (CV+PG+F).



841

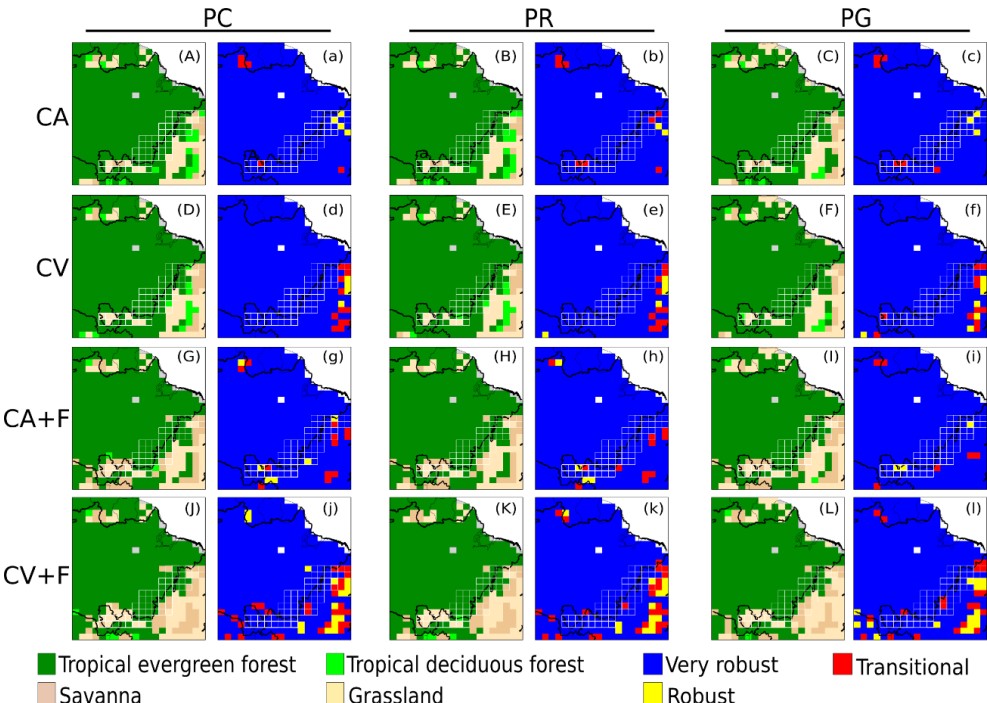

842

**Figure 6.** Results for the dominant vegetation cover simulated by INLAND for the different treatments

(A-L) and a metric of variability of results (a-l). Simulations are considered very robust if the dominant

vegetation agrees on 9-10 of the last 10 years of simulation, robust if it agrees on 7-8 years, and

transitional if on 6 or fewer years.





**Table 1.** Simulations with different scenarios evaluated by INLAND model in Amazonia-Cerrado
transition. CA, climatological average, 1961-1990; CV, monthly climate data, 1948-2008; the nutrient
limitation on $V_{max}$ - PC, no phosphorus limitation ($V_{max} = 65$ µmolCO$_2$ m$^{-2}$ s$^{-1}$); PR, regional phosphorus
limitation; PG, global phosphorus limitation).

| Climate | CO$_2$ | Fire (F) | $V_{max}$ | | |
|---|---|---|---|---|---|
| | | | PC | PR | PG |
| CA | Variable | Off | CA+PC | CA+PR | CA+PG |
| CA | Variable | On | CA+PC+F | CA+PR+F | CA+PG+F |
| CV | Variable | Off | CV+PC | CV+PR | CV+PG |
| CV | Variable | On | CV+PC+F | CV+PR+F | CV+PG+F |



**Table 2.** Individual and combined effects for each simulation in Amazonia-Cerrado transition. CA,
climatological seasonal average, 1961-1990; CV, monthly climate data, 1948-2008; the nutrient limitation
on $V_{max}$ - PC, no phosphorus limitation ($V_{max} = 65$ µmolCO$_2$ m$^{-2}$ s$^{-1}$); PR, regional phosphorus limitation;
PG, global phosphorus limitation)

| Climate (C) | Phosphorus (P) | Fire (F) |
|---|---|---|
| (CV+PC)-(CA+PC) | (CA+PR)-(CA+PC) | (CA+PC+F)-(CA+PC) |
| (CV+PR)-(CA+PR) | (CV+PR)-(CV+PC) | (CV+PC+F)-(CV+PC) |
| (CV+PG)-(CA+PG) | (CA+PG)-(CA+PC) | (CA+PR+F)-(CA+PR) |
| | (CV+PG)-(CV+PC) | (CV+PR+F)-(CV+PR) |
| | | (CA+PG+F)-(CA+PG) |
| | | (CV+PG+F)-(CV+PG) |






**Table 3.** Summary of average NPP, LAI and AGB for the Amazonia-Cerrado transition at the
transects domains, considering all simulations with CA and CV regardless of fire presence or phosphorus
limitation. The results of a one-way ANOVA are also shown, including the *F* statistic, and p value. Values
within each column followed by a different letter are significantly different (p < 0.05) according to the
Tukey–Kramer test.

| Group 1 | **NPP** $\text{kg-C m}^{-2}\text{ yr}^{-1}$ | | **LAI$_{total}$** $\text{m}^2\text{ m}^{-2}$ | | **LAI$_{lower}$** $\text{m}^2\text{ m}^{-2}$ | | **LAI$_{upper}$** $\text{m}^2\text{ m}^{-2}$ | | **AGB** $\text{kg-C m}^{-2}$ | |
|---|---|---|---|---|---|---|---|---|---|---|
| CA | 0.68 | a | 7.47 | a | 1.98 | a | 5.49 | a | 6.68 | a |
| CV | 0.64 | b | 7.15 | b | 2.11 | a | 5.04 | b | 6.30 | b |
| *F* | *40.2* | | *57.2* | | *2.96* | | *36.0* | | *11.3* | |
| *p* | *<0.001* | | *<0.001* | | *ns* | | *<0.01* | | *<0.001* | |


**Table 4.** Summary of average NPP, LAI and AGB for the transition at the transects domains, considering
different phosphorus limitation, regardless of climate and fire presence. The results of a one-way ANOVA
are also shown, including the *F* statistic, and p value. Values within each column followed by a different
letter are significantly different (p < 0.05) according to the Tukey–Kramer test.

| Group 2 | **NPP** $\text{kg-C m}^{-2}\text{ yr}^{-1}$ | | **LAI$_{total}$** $\text{m}^2\text{ m}^{-2}$ | | **LAI$_{lower}$** $\text{m}^2\text{ m}^{-2}$ | | **LAI$_{upper}$** $\text{m}^2\text{ m}^{-2}$ | | **AGB** $\text{kg-C m}^{-2}$ | |
|---|---|---|---|---|---|---|---|---|---|---|
| PC | 0.71 | a | 7.64 | a | 1.84 | b | 5.80 | a | 7.15 | a |
| PR | 0.64 | b | 7.15 | b | 2.19 | a | 4.95 | b | 6.20 | b |
| PG | 0.64 | b | 7.14 | b | 2.10 | a | 5.04 | b | 6.12 | b |
| *F$_{2.99}$* | *62.8* | | *61.0* | | *8.75* | | *53.5* | | *33.6* | |
| *p* | *<0.001* | | *<0.001* | | *<0.01* | | *<0.01* | | *<0.001* | |



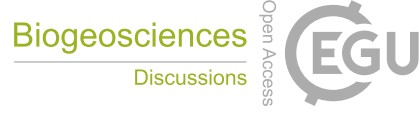

**Table 5.** Summary of average NPP, LAI and biomass for the transition at the transects domains,
considering presence or absence of fire. The results of a one-way ANOVA are also shown, including the
*F* statistic, and p value. Values within each column followed by a different letter are significantly different
(p < 0.05) according to the Tukey–Kramer test.

| Group 3 | NPP | | LAI$_{total}$ | | LAI$_{lower}$ | | LAI$_{upper}$ | | AGB | |
|---|---|---|---|---|---|---|---|---|---|---|
| | kg-C m$^{-2}$ yr$^{-1}$ | | m$^2$ m$^{-2}$ | | m$^2$ m$^{-2}$ | | m$^2$ m$^{-2}$ | | kg-C m$^{-2}$ | |
| Fire OFF | 0.66 | a | 6.72 | b | 0.88 | b | 5.84 | a | 8.47 | b |
| Fire ON | 0.67 | b | 7.90 | a | 3.21 | a | 4.69 | b | 4.51 | a |
| $F_{3.84}$ | 8.28 | | 937 | | 1459 | | 249 | | 1719 | |
| $p$ | <0.005 | | <0.001 | | <0.01 | | <0.01 | | <0.001 | |





**Table 6.** Summary of average NPP, LAI and AGB for the transition at the transects domains, considering all factor combinations. The results of a one-way ANOVA are also shown, including the *F* statistic, and p value. Values within each column followed by a different letter are significantly different (p < 0.05) according to the Tukey–Kramer test.

| | NPP | | LAI$_{total}$ | | LAI$_{lower}$ | | LAI$_{upper}$ | | AGB | |
|---|---|---|---|---|---|---|---|---|---|---|
| | kg-C m$^{-2}$ yr$^{-1}$ | | m$^2$ m$^{-2}$ | | m$^2$ m$^{-2}$ | | m$^2$ m$^{-2}$ | | kg-C m$^{-2}$ | |
| CV+PC | 0.69 | bcd | 6.96 | d | 0.84 | e | 6.48 | a | 9.01 | ab |
| CV+PG | 0.61 | f | 6.24 | f | 0.85 | e | 5.60 | bc | 7.91 | c |
| CV+PR | 0.62 | f | 6.33 | f | 0.85 | e | 5.74 | bc | 8.04 | c |
| CV+PC+F | 0.69 | abc | 7.92 | b | 2.91 | cd | 4.61 | ef | 4.89 | de |
| CV+PG+F | 0.63 | ef | 7.76 | b | 3.73 | a | 5.81 | bc | 3.91 | f |
| CV+PR+F | 0.63 | ef | 7.65 | bc | 3.47 | ab | 4.69 | ef | 4.02 | f |
| CA+PC | 0.72 | ab | 7.39 | c | 0.91 | e | 6.12 | ab | 9.31 | a |
| CA+PG | 0.64 | def | 6.64 | e | 0.91 | e | 5.40 | cd | 8.22 | c |
| CA+PR | 0.65 | cdef | 6.72 | de | 0.91 | e | 5.49 | cd | 8.31 | bc |
| CA+PC+F | 0.74 | a | 8.29 | a | 2.69 | d | 5.02 | de | 5.40 | d |
| CA+PG+F | 0.67 | cde | 7.90 | b | 3.29 | abc | 4.04 | g | 4.45 | ef |
| CA+PR+F | 0.67 | cde | 7.88 | b | 3.19 | bc | 4.18 | fg | 4.42 | ef |
| *F* | *16.2* | | *115* | | *140* | | 38.1 | | *172* | |
| *p* | *<0.001* | | *<0.001* | | *<0.01* | | *<0.01* | | *<0.001* | |





**Table 7**. Correlation coefficients of biomass simulated by INLAND and field estimates

|          | T1    | T2    | T3    | T4    | T5    | All transects |
|----------|-------|-------|-------|-------|-------|---------------|
| CA+PC    | 0.843 | 0.928 | 0.886 | 0.937 | 0.337 | 0.786 |
| CV+PC    | 0.838 | 0.884 | 0.890 | 0.939 | 0.355 | 0.781 |
| CA+PR    | 0.793 | 0.848 | 0.830 | 0.911 | 0.399 | 0.756 |
| CV+PR    | 0.795 | 0.793 | 0.832 | 0.907 | 0.527 | 0.771 |
| CA+PG    | 0.814 | 0.951 | 0.838 | 0.889 | 0.388 | 0.776 |
| CV+PG    | 0.825 | 0.922 | 0.840 | 0.879 | 0.496 | 0.792 |
| CA+PC+F  | 0.988 | 0.987 | 0.977 | 0.892 | 0.133 | 0.795 |
| CV+PC+F  | 0.976 | 0.947 | 0.933 | 0.908 | 0.187 | 0.790 |
| CA+PR+F  | 0.842 | 0.805 | 0.981 | 0.808 | 0.561 | 0.799 |
| CV+PR+F  | 0.925 | 0.804 | 0.927 | 0.808 | 0.319 | 0.757 |
| CA+PG+F  | 0.844 | 0.961 | 0.980 | 0.830 | 0.430 | 0.809 |
| CV+PG+F  | 0.845 | 0.932 | 0.931 | 0.881 | 0.177 | 0.753 |
| CA avg   | 0.854 | 0.913 | 0.915 | 0.878 | 0.375 | 0.787 |
| CV avg   | 0.867 | 0.880 | 0.892 | 0.887 | 0.344 | 0.774 |


