# Peer review of "Influence of climate variability, fire and phosphorus limitation on the vegetation structure and dynamics in the Amazon-Cerrado border"

_Biogeosciences, 2016_

## Referee Comment (RC1) · Anonymous Referee #1 · 2 Feb 2017

General comments: This manuscript investigates the drivers of the savanna-forest border in South America. The authors examine the effects of climatic variability by forcing the model using two different datasets, of phosphorus limitation by altering Vmax based on two different phosphorus datasets and by running simulations with fire on and fire off.

The manuscript attempts to answer the long standing question about the actual drivers of the cerrado-amazon biome boundary and is the first to test the effects of phosphorus limitation in the cerrado. The simulation results for the Amazon region using the

regional phosphorus map (PR) are very similar to those produced by Castanho et al. (2013) given it is the same data and essentially the same model. The comparison between the global and regional phosphorus map is interesting as is its extension into the cerrado. It would be interesting to see which most closely matched satellite derived biomass data (Saatchi/Baccini/Avitabil) for the entire study area, my guess is that the PR simulations would – you talk about the biomass distributions across the Amazon region anyway, it would be useful if the reader could visualise this in some way.

There is one aspect of the manuscript/model which causes me particular concern, how do you simulate such low biomass and predominantly grasslands and savanna in the cerrado area with fire turned off? Precipitation ranges between ca. 1000 & 2000 mm/yr in this area (maps of your forcing data would remove the need to guess the precipitation range which generated these results). It is generally accepted that above ca. 800 mm/yr, fire (or some other limiting factor) is necessary to prevent the formation of closed canopy forest/woodland vegetation formations (e.g. Hoffmann et al., 2012). Your results (Fig. 6) however show that, in an area where precipitation is well above this threshold, neither phosphorus limitation nor fire are necessary to explain the presence of what looks to be about 65% of the distribution of c4 dominated vegetation formations in the cerrado. This result is incredible; you need to explain how/why your model behaves like this. The result contradicts most of the savanna ecological literature and needs to be discussed and justified in detail.

Overall the manuscript presents novel, interesting results and a potentially new (but not discussed) perspective on the drivers of mesic savanna distributions. The manuscript is let-down by the presentation quality to such an extent that it makes it difficult to assess the scientific significance of the work; the text needs careful re-editing, methods need to be more detailed to allow the assessment of their validity, the presented but not discussed new perspective on the drivers of mesic c4 dominated vegetation formations needs particular attention.

Specific comments: The manuscript needs to be carefully edited to improve English,
currently it is difficult to understand and cumbersome to read. In many instances this does not detract from the message however there are places where I cannot understand what the authors are trying to convey.

I would be reluctant to conclude that inter-annual climate variability in general does not play a role in determining the transition. Also, sometimes you write climate variability and sometimes inter-annual climate variability, these are two very different things; make it clear that when you refer to climate variability you are actually talking about the difference between two datasets used to force your model. How different are the CA and CV data? Some plots in the supplementary materials would be useful.

I would like you to state how you calculate your sample sizes, how big your sample sizes are, their means and standard deviations for all of your statistical tests. This is missing from the methods which makes it difficult to assess the appropriateness of the statistics used.

I'm quite impressed that your biomass falls from west to east (Fig. 5 T1-T4) in the absence of fire, it appears that your simulated biomass responds well to reductions in precipitation /& increased dry season length. Other models don't appear to respond this well, see Fig.3 in Galbraith et al. (2010). Looking at the biomass produced by IBIS (INLAND is based on this) for the cerrdo (Plate 2.b in Foley et al. (1996)) it would appear that the biomass you are simulating is much lower than that presented by Foley.

The simulated biome distributions are excellent, however, I'm very surprised to see such large savanna and grassland extents in the absence of fire (Fig. 6). Most models would simulate relatively high biomass tropical evergreen or deciduous forest in the absence of fire, in fact, most models simulate relatively high biomass tropical evergreen or deciduous forest in the presence of fire (e.g. Fig. 4 in Bond et al. (2005), Fig. 2 in Smith et al., (2014) and Plate 7 in Foley et al. (1996) – INLAND is based on IBIS so I'm wondering why there is such a big difference). Based on your results you are simulating savanna and grassland through most of the cerrado (these are also "Very robust") with

BGD
fire turned off, does this mean the presence of the cerrado/c4 dominated vegetation does not depend on fire? How/why does the model simulate these vegetation types in areas with such high precipitation in the absence of fire? Has the model been re-tuned, if so how?

Technical corrections: 1. There are many grammatical errors and confusing sentences throughout the manuscript. 2. T5 missing from Fig. 1. 3. It could be made more clear what Fig. 3b is showing. 4. It is incredibly difficult to distinguish the different greys used in Fig. 5. 5. Line 502: T2 and T3 show the highest average correlations not T3 & T4. Following this it is stated that biomass in these transects are (Fig. 5 b & d) are underestimated due to lower water availability. T2, T3 and T4 mostly overestimate, not underestimate, biomass, apart from the simulations with fire in which case it would be due to the presence of fire? Additionally, it would be useful if it was indicated in Fig. 5 somewhere which points are cerrado and which are forest. The text refers repeatedly to cerrado and forest points but I can't tell which are which from the figure and need to constantly refer to Fig. 1. See also my greyscale comment. 6. I would recommend leaving the discussion to the discussion section to remove repetition (e.g. line 272 "reinforcing . . ..", line 311 "which is relevant") however this can be very difficult. 7. Line 661 – the author list repeats itself.

References Bond, W. J., Woodward, F. I. and Midgley, G. F.: The global distribution of ecosystems in a world without fire, New Phytol., 165(2), 525–538, doi:10.1111/j.1469-8137.2004.01252.x, 2005. Castanho, A. D. A., Coe, M. T., Costa, M. H., Malhi, Y., Galbraith, D. and Quesada, C. A.: Improving simulated Amazon forest biomass and productivity by including spatial variation in biophysical parameters, Biogeosciences, 10(4), 2255–2272, doi:10.5194/bg-10-2255-2013, 2013. Foley, J. A., Prentice, I. C., Ramankutty, N., Levis, S., Pollard, D., Sitch, S. and Haxeltine, A.: An integrated biosphere model of land surface processes, terrestrial carbon balance, and vegetation dynamics, Glob. Biogeochem. Cycles, 10(4), 603–628, doi:10.1029/96GB02692, 1996. Galbraith, D., Levy, P. E., Sitch, S., Huntingford, C., Cox, P., Williams, M.
and Meir, P.: Multiple mechanisms of Amazonian forest biomass losses in three dynamic global vegetation models under climate change, New Phytol., 187(3), 647– 665, doi:10.1111/j.1469-8137.2010.03350.x, 2010. Hoffmann, W. A., Geiger, E. L., Gotsch, S. G., Rossatto, D. R., Silva, L. C. R., Lau, O. L., Haridasan, M. and Franco, A. C.: Ecological thresholds at the savanna-forest boundary: how plant traits, resources and fire govern the distribution of tropical biomes, Ecol. Lett., 15(7), 759–768, doi:10.1111/j.1461-0248.2012.01789.x, 2012. Smith, B., W\a arlind, D., Arneth, A., Hickler, T., Leadley, P., Siltberg, J. and Zaehle, S.: Implications of incorporating N cycling and N limitations on primary production in an individual-based dynamic vegetation model, Biogeosciences, 11(7), 2027–2054, doi:10.5194/bg-11-2027-2014, 2014.

---

## Referee Comment (RC2) · Anonymous Referee #2 · 3 Feb 2017

This study analyses the INLAND vegetation model with the purpose of discerning the relative impacts of fire, empirical phosphorus limitation and climate variability on predictions of ecosystem structure across forest-cerrado transitions in S. America.

In common with reviewer #1, I think that the text requires careful editing, particularly for (mostly minor but widespread) grammatical errors.

The model description is extremely vague, and and parameterization and calibration carried out prior to these experiments is omitted. I am skeptical that the model simply performed reasonably the first time that it was run. What uncertainties do you need

to grapple with before the model output falls within the sensible range? Without this information, the reader might assume that goodness-of-fit tests between the models and the observations might have been substantially affected by undisclosed model calibration.

Given that, I find the comparison of different influences over model outputs (fire, phosphorus, etc.) to be somewhat predictable and not very interesting. The reliance on statistical tests is distracting. A better analysis of the consequences of and the uncertainty in the impacts would be much more useful.

The discussion section contains numerous logical errors confusing the output of the model and the behaviour of the ecosystem in real life. Until these are rectified, I do not think that this paper is of sufficiently high scientific standard to be published.

Specific Comments.

L112: Is Kucharik (200) really the most recent reference for the INLAND model? I'm fairly sure this isn't the case. To be repeatable, this model description needs to provide at a minimum references to the most recent version, along with specific descriptions of the model equations and parameters if they have been modified since the last publication. Many EGU journals stipulate that directions to the code used are also included. I do not know if this applies to BGD, but it would be good practice to do so.

L115: Does this mean the vegetation types compete for light, or for water & nutrients? The mechanisms of competition and dynamic vegetation are a critical part of a model of this type. I am surprised you skipped over this so briefly.

L116: This classification seems arbitrary to me. Why not just report the LAI numbers?

L122: Again, I'm not sure of the need for this cross-referencing of PFTs, 'vegetation types' (why not ecosystem type - that would be less confusing) and then names for the ecosystems. The purpose of a mechanistic model is to describe the system quantitatively and in multiple dimensions. Introducing a simplistic written classification scheme

does not seem. to add any extra information.

L129: How is it similar to Century and how is it different? Small differences can be important in dynamical non-linear models.

L136: From where did this relationship between P and Vcmax arise? Is it sensible to use is across this biome? More detail is needed in addition to giving the reference, given how central this relationship is to the rest of the analysis.

L140: Again, how it is similar to CTEM? How dos the arbitrary ignition scheme work? If this is covered later in the text, it should be referenced here.

L145: Why bring up the two options if only CTEM is used in this study?

L185-192: This description seems more like a discussion than methods. Also, can you clarify the impact of land use on these transitions?

L194-197: I'm not sure what point you are trying to make here.

L212: This description of the model experiments needs cleaning up. Only the P limitation scenarios seems to have label (PC, PR, etc.) and what the combinations are is not discussed at all in the text, nor are the number of scenarios, etc.

L220: It is not yet clear how the model distinguishes upper and lower canopy LAI? Therefore this distinction is not useful yet as a diagnostic.

L235: Given these are deterministic model outputs, why conduct these statistic tests? Only one instance of the atmospheric forcing, boundary conditions, parameters, and model structure is sampled, so what does it tell you if the difference between one model run or another is 'significant'? This might make sense if applied to ensembles of runs, but to compare one run against another it seems inappropriate.

L275: This is over-stating the conclusions of the model. No real evidence is presented here that it correctly simulates the complex biophysics of forest flammability, so to draw this conclusion (that the model 'shows that the Amazonia(n) forest is naturally

inflammable) is not defensible.

L279: It seems strange to me that, in the absence of a detailed illustration that the model functions appropriately in these regions, there is no investigation of any type of within-model variability, and the structure and parameterizations of the model are assumed to be fixed. I see that this study aims to look at large modifications in model scope, but I find it unusual that no other model features are brought into question at all, particularly with regard to the strength of the conclusions.

L309: My reading of figure 5 is that the full model, with all elements, under-predicts biomass significantly over much of the transition region (transect 1 and 4 in particular). Table 7 only presents correlations and not biases, so this feature is glossed over.

L313: How does this finding relate to those of the Senna and Castanho studies? This is too vague a reference.

L320: This is a highly complex system and biases can and do arise from a huge number of sources. It is not necessarily a local problem, nor anything to do with moisture stress - those are both unfounded speculations.

L337: These conclusions - that phosphorus limitation and fire tend to reduce vegetation biomass, are pretty self-evident and not very interesting.

L339: The word 'robust' here is problematic. The model does not show deviate through time in these fields, but 'robust' is normally used to describe a simulation which is physically plausible, and I don't think that applies here necessarily.

L363: Again, changing the model drastically 'led to significantly different average biomass' is not a very interesting conclusion from a piece of science. I do not think there is any debate about whether fire reduces forest biomass where it occurs, nor whether introducing a universally lower Vcmax might reduce vegetation productivity?

L369: Is climatic inter-annual variability in the CRU dataset realistic? There are other climate reanalyses that one might test it against, as well as station-level meteorological

records. Given the incompleteness of met station data across this domain, it would seem likely that it underestimates variability somewhat.

L372: This is a very old reference for this very active field.

L413: How is the adaptation of savanna species to P-limitation represented in the model? As far as I can tell, the impact of P on Vcmax was universal and not PFT specific?

L426: These outputs do not actually show that understanding phosphorus limitation is associated with reliability of databases, it just shows that the databases are different. An alternative model structure that is not so sensitive to overall soil P, for example, might conclude that the database discrepancy doesn't matter? That is a hypothetical example, but the logic of this sentence is unconvincing.

L432: References to the state of the art in nutrient cycle modeling should be included here.

L444: This logic (it is clear that phosphorus has a significant effect on woody biomass) is also unconvincing. It simply shows that the (simplistic) model predicts this, not that it happens in real life.

L447: Again, this simply shows that this fire model does not burn the intact forest, and this cannot be used to conclusively state anything about real intact forest.

L459: Are the physiological differences between cerrado and other vegetation types depicted in the CTEM model? Again, the sparse model description does not allow this to be determined.

L491: Given that there is no indication of how the parameterization for rainforest vegetation came to be in the first instance, one cannot say whether the P limitation should necessarily be an improvement. LAI in biosphere models can be modified trivially by changing the leaf lifespan and/or specific leaf area or leaf allocation scheme. All of these features are variable in observational datasets, and so the initial LAI predictions

can, I am pretty certain, be modified massively. Whether the model over or under-predicts LAI in the first instance is therefore a matter of parameter choice, and therefore whether the P limitation improves or degrades the model is also a feature of that choice.

L519: You predict that the vegetation distribution is affected by these things, not observe.

L540: This is an extraordinarily grandiose and unneeded claim. I'm pretty sure that, for example, Levine et al. (2016) might disagree.

L555: Bringing up the need for greater constraint on the uncertain model parameters at this point seems a bit too-little too-late.

Figure 1) I don't see how the transects are dilineated in this figure?

Figure 2) Definition of 'new' is ambiguous in the legend. As is the use of the '-' sign to denote PR and PG. Unclear if it means 'minus' or not.

Figure 3) What is figure b? It doesn't say in the legend.

Table 7: Why only correlation coefficients and not also biases?

Levine, Naomi M., et al. "Ecosystem heterogeneity determines the ecological resilience of the Amazon to climate change." Proceedings of the National Academy of Sciences 113.3 (2016): 793-797.

---

## Author Comment (AC1) · 24 Mar 2017

Response to Reviewer on Paper doi:10.5194/bg-2016-510

(Reviewer comments in *italics;* Responses in **bold**)

Response to Anonymous Referee #1

**We are grateful to the reviewers for their insightful comments and helpful suggestions. Please find detailed responses to each comment below.**

*General comments*
*This manuscript investigates the drivers of the savanna-forest border in South America. The authors examine the effects of climatic variability by forcing the model using two different datasets, of phosphorus limitation by altering Vmax based on two different phosphorus datasets and by running simulations with fire on and fire off.*
*The manuscript attempts to answer the long standing question about the actual drivers of the cerrado-amazon biome boundary and is the first to test the effects of phosphorus limitation in the cerrado. The simulation results for the Amazon region using the regional phosphorus map (PR) are very similar to those produced by Castanho et al. (2013) given it is the same data and essentially the same model. The comparison between the global and regional phosphorus map is interesting as is its extension into the cerrado. It would be interesting to see which most closely matched satellite derived biomass data (Saatchi/Baccini/Avitabil) for the entire study area, my guess is that the PR simulations would – you talk about the biomass distributions across the Amazon region anyway, it would be useful if the reader could visualise this in some way.*

**Response: Our focus is the evaluation of the transition region between Amazonia and Cerrado, where many physiognomic mosaics exist. The main reason why we don´t use Saatchi, Baccini, and Avitabil data is that they are actual maps of biomass, including data from deforested pixels, which are very common in the region. Since we are simulating natural vegetation, such comparison would be misleading. We prefer to use the biomass database of Nogueira et al. (2015) that includes aboveground measures in the original vegetation (before extensive clearing), and uses 74 different classes of vegetation, and thus represents better the many physiognomies in the region. The areas that are currently degraded were identified, and biomass was assigned according to information about the original vegetation.**

*There is one aspect of the manuscript/model which causes me particular concern, how do you simulate such low biomass and predominantly grasslands and savanna in the cerrado area with fire turned off? Precipitation ranges between ca. 1000 & 2000 mm/yr in this area (maps of your forcing data would remove the need to guess the precipitation range which generated these results). It is generally accepted that above ca. 800 mm/yr, fire (or some other limiting factor) is necessary to prevent the formation of closed canopy*

*forest/woodland vegetation formations (e.g. Hoffmann et al., 2012). Your results (Fig. 6) however show that, in an area where precipitation is well above this threshold, neither phosphorus limitation nor fire are necessary to explain the presence of what looks to be about 65% of the distribution of c4 dominated vegetation formations in the cerrado. This result is incredible; you need to explain how/why your model behaves like this. The result contradicts most of the savanna ecological literature and needs to be discussed and justified in detail.*

**Response: The focus on annual mean precipitation is an oversimplification of the climate controls on vegetation dynamics, in particular in hot tropical regions. In INLAND, photosynthesis does not depend only on water availability, but also on hourly temperatures, atmospheric $CO_2$ concentration and water vapor pressure. The high temperatures (easily above 35°C during the dry season), and the water stress are factors sufficient to reduce net primary production to a level compatible with the simulation of savanna and grass vegetation classes in the southeast portion of the simulated area, even in the absence of further disturbances such as fire. These classes, however, have somewhat high biomass, indicating the need of additional factors increasing mortality or reducing growth, namely fire and phosphorus limitation.**

**Using the LAI criteria for the dominant plant functional type our simulations show that, in the absence of fire, the border region between the Amazon and Cerrado is dominated by the presence of tropical evergreen forest, when in fact they should show the presence of deciduous forest or savannas. In addition, the presence of grasslands and savannas are restricted to the southeastern portion of the simulated area, in response to the higher water stress (Figure 6ABCDEF).**

**Botta and Foley (2002) (see their Figure 2b, appended below), using IBIS, found similar results to simulate vegetation distribution considering CRU climatology and absence of disturbances such as fire or nutritional limitation, showing that IBIS has a sensitivity to seasonal variability of water availability in the southeast of the Amazon basin.**

[Figure]

**Figure 2.** Potential vegetation map (dark green: evergreen broadleaf forest, light green: deciduous broadleaf forest, blue: evergreen needleleaf forest, orange: savanna, yellow: grassland/steppe, red: shrubland, black: desert/semidesert) (a) adapted from *Ramankutty and Foley* [1999] and (b) simulated by IBIS with no disturbance and mean climate.

Source: Botta, A. and Foley, J. A.: Effects of climate variability and disturbances on the Amazonian 590 terrestrial ecosystems dynamics, Global Biogeochem. Cycles, 16(4), doi:10.1029/2000GB001338, 2002.

*Overall the manuscript presents novel, interesting results and a potentially new (but not discussed) perspective on the drivers of mesic savanna distributions. The manuscript is let-down by the presentation quality to such an extent that it makes it difficult to assess the scientific significance of the work; the text needs careful re-editing, methods need to be more detailed to allow the assessment of their validity, the presented but not discussed new perspective on the drivers of mesic c4 dominated vegetation formations needs particular attention.*

**Response: The manuscript has been completely rewritten. We hope it will satisfy the reviewer now.**

*Specific comments:*

*The manuscript needs to be carefully edited to improve English, currently it is difficult to understand and cumbersome to read. In many instances this does not detract from the message however there are places where I cannot understand what the authors are trying to convey.*

**Response: We apologize for the mistakes. The manuscript has been completely rewritten. We hope it will satisfy the reviewer now.**

*I would be reluctant to conclude that inter-annual climate variability in general does not play a role in determining the transition (1).*

**Response: Part of our analyses aimed at quantifying the importance of the individual drivers, using statistical analyses. Our results demonstrate that the inter-annual climate variability effect has lower magnitude than phosphorus limitation and fire on aboveground biomass in the analyzed pixels. This does not mean that inter-annual climate variability does not play a role along the transition, but in the 31 pixels along the transition the effect of CV is smaller than the other factors. Our spatial analysis (Figure 4a) demonstrate that CV has significant effect on the level of 95% according to student's t-test, especially on the northeast and southeast of simulated domain favoring occurrence of grasslands and savanna in the Cerrado domain.**

*Also, sometimes you write climate variability and sometimes inter-annual climate variability, these are two very different things; make it clear that when you refer to climate variability you are actually talking about the difference between two datasets used to force your model(2).*

**Response: We meant to use climate variability and inter-annual climate variability interchangeably, but with the same meaning. We have now changed the revised manuscript to make it more clear.**

*How different are the CA and CV data? Some plots in the supplementary materials would be useful (3).*

**Response: The main difference between CA and CV is own the inter-annual climate variability. While in CA simulations we repeat the same climate conditions (average for 1960-1990), in CV the set of 61 years of inter-annual climate variability was used. We plotted the climate average (1960-1990) and the last ten years of the set inter-annual climate variability (1998-2008) for Cerrado and Amazon biomes (Figure A).**

The Figure A shows that the range of seasonality of precipitation in CA databases for Amazon and Cerrado is lower than CV, and that the rainfall in wet months generally is higher. For the dry season, however, the average of CV precipitation in Amazonia is higher than CA for most part of the years (Figure Ab), and lower for Cerrado.

[Figure]

**Figure A. Climate average (1960-1990) and inter-annual climate variability (1998-2008) for Amazon and Cerrado biomes.**

*I would like you to state how you calculate your sample sizes, how big your sample sizes are, their means and standard deviations for all of your statistical tests. This is missing from the methods which makes it difficult to assess the appropriateness of the statistics used.*

**Response: In the spatial analysis we use the last 10 years of simulation (to make sure simulations are in near-equilibrium, reflecting the effects of the latest $CO_2$), to test the differences between the treatments by t-test at $p<0.05$. The results are tested in each pixel, for all the simulated domain (n = 10).**
**For the transects we present an analysis of variance using the one-way ANOVA and the Tukey-Kramer test. We considered the 31 pixels corresponding to transects from T1 to T5 ($n_{pixel}$), the set of last 10 years of simulation (1999 - 2008) ($n_{year}$) and the 12 simulations ($n_{simulation}$) (Table 1). We grouped treatments according to climate, where all sets with CV vs CA are tested (Group 1, n=1860, ($n_{pixel}$ x $n_{year}$ x $n_{simulation/2}$); nutrient limitation, PC, PR and PG are tested regardless the F or climate used (Group 2, n=1240, ($n_{pixel}$ x $n_{year}$ x $n_{simulation/3}$); and presence or absence of fire regardless the F or P presence (Group 3, n=1860, ($n_{pixel}$ x $n_{year}$ x $n_{simulation/2}$). We are providing the inclusion of this information and the means and standard deviations on the manuscript. Please check section 2.5.**

*I'm quite impressed that your biomass falls from west to east (Fig. 5 T1-T4) in the absence of fire, it appears that your simulated biomass responds well to reductions in precipitation /& increased dry season length. Other models don't appear to respond this well, see Fig.3 in Galbraith et al. (2010). Looking at the biomass produced by IBIS (INLAND is based on this) for the cerrado (Plate 2.b in Foley et al. (1996)) it would appear that the biomass you are simulating is much lower than that presented by Foley.*
*The simulated biome distributions are excellent, however, I'm very surprised to see such large savanna and grassland extents in the absence of fire (Fig. 6). Most models would simulate relatively high biomass tropical evergreen or deciduous forest in the absence of fire, in fact, most models simulate relatively high biomass tropical evergreen or deciduous forest in the presence of fire (e.g. Fig. 4 in Bond et al. (2005), Fig. 2 in Smith et al., (2014) and Plate 7 in Foley et al. (1996) – INLAND is based on IBIS so I'm wondering why there is such a big difference).*

**Response: Two considerations must be made here. The first is that the model used by Foley et al. (1996) is the first version of IBIS, and is not the best one to compare to. After that, other implementations and modifications in the code were made. In fact, INLAND is based on the IBIS version used by Senna et al. (2009), after which no major changes have been made to the calibration. Second, we also need to consider the differences in the code, boundary conditions, and observed datasets used in these studies.**

*Based on your results you are simulating savanna and grassland through most of the cerrado (these are also "Very robust") with fire turned off, does this mean the presence of the cerrado/c4 dominated vegetation does not depend on fire? How/why does the model simulate these vegetation types in areas with such high precipitation in the absence of fire? Has the model been retuned, if so how?*

**Response: The vegetation simulated by INLAND in regions with long dry season, is characterized by rare perennial trees and deciduous trees dominate the upper canopy. Tree NPP is low, and trees have a relatively low LAI, allowing light to penetrate in the canopy, allowing development of grasses. Mathematically, it is possible to simulate a low biomass vegetation in a tropical seasonal climate either by adding disturbance, or by having the correct response of NPP to water stress (see equation below).**

$$\frac{dB}{dt} = a \cdot NPP - m \cdot B - d \cdot B$$

**Where B is biomass, a is the allocation coefficient, m is natural mortality, d is disturbance mortality. For a near equilibrium case, for a and m fixed, B can be in the range of savanna biomass if d is large and NPP is high, or if d is low and NPP is low, or if d = 0 and NPP is lower. All these cases lead to the biomass in the correct savanna range.**

**What we argue is that in a few pixels it is possible to have savanna/grass as dominant vegetation without the presence of fire, just because the tree NPP is low enough. However, this distribution does not represent the real Amazon-Cerrado limits. This distribution is further improved after fire is enabled and NPP is reduced due to phosphorus.**

**Regarding tuning, the latest tuning of the model was described by Senna et al. 2009, which is the version of IBIS that led to INLAND.**

*Technical corrections:*
*1. There are many grammatical errors and confusing sentences throughout the manuscript.*
**Response:  We apologize for the mistakes. A revised version of manuscript was uploaded. We hope it will satisfy the reviewer now.**

*2. T5 missing from Fig. 1.*
**Response: This has been changed.**

[Figure]

**Figure 1.**

*3. It could be made more clear what Fig. 3b is showing.*
**Response: This information was added.**

*4. It is incredibly difficult to distinguish the different greys used in Fig. 5.*
**Response: This has been changed. The revised manuscript includes a new Figure.**

[Figure]

**Figure 5.**

*5. Line 502: (A) T2 and T3 show the highest average correlations not T3 & T4. Following this it is stated that biomass in these transects are (Fig. 5 b & d) are underestimated due to lower water availability. T2, T3 and T4 mostly overestimate, not underestimate, biomass, apart from the simulations with fire in which case it would be due to the presence of fire?*

*(B)Additionally, it would be useful if it was indicated in Fig. 5 somewhere which points are cerrado and which are forest. The text refers repeatedly to cerrado and forest points but I can't tell which are which from the figure and need to constantly refer to Fig. 1. See also my greyscale comment.*

*Response:*
**(A)We thank for this observation. This error has been changed in the text.**
**(B) This has been changed.**

*6. I would recommend leaving the discussion to the discussion section to remove repetition (e.g. line 272 "reinforcing . . ..", line 311 "which is relevant") however this can be very difficult. 7.*
**Response: This has been changed.**

*Line 661 – the author list repeats itself.*
**Response: This has been changed.**

**References**

Baccini, A., Goetz, S. J., Walker, W. S., Laporte, N. T., Sun, M., Sulla-Menashe, D., Hackler, J., Beck, P. S. A., Dubayah, R., Friedl, M. A., Samanta, S., and Houghton, R. A.: Estimated carbon dioxide emissions from tropical deforestation improved by carbon-density maps, Nat. Climate Change, 2, 182– 185; doi:10.1038/nclimate1354, 2012.

Botta, A. and Foley, J. A.: Effects of climate variability and disturbances on the Amazonian 590 terrestrial ecosystems dynamics, Global Biogeochem. Cycles, 16(4), doi:10.1029/2000GB001338, 2002.

Foley, J. A., Prentice, I. C., Ramankutty, N., Levis, S., Pollard, D., Sitch, S. and Haxeltine, A.: An integrated biosphere model of land surface processes, terrestrial carbon balance, and vegetation dynamics, Glob. Biogeochem. Cycles, 10(4), 603–628, doi:10.1029/96GB02692,
1996.

Nogueira, E. M., Yanai, A. M., Fonseca, F. O. and Fearnside, P. M.: Carbon stock loss from 705 deforestation through 2013 in Brazilian Amazonia, Glob. Chang. Biol., doi:10.1111/gcb.12798, 2015.

Senna, M. C. A., Costa, M. H., Chipponelli Pinto, L. I., Acioli Imbuzeiro, H. M., Freitas Diniz, L. M. and Pires, G. F.: Challenges to reproduce vegetation structure and dynamics in Amazonia using a coupled climate-biosphere model, Earth Interact., 13(11), doi:10.1175/2009EI281.1, 2009.

Saatchi, S. S., Harris, N. L., Brown, S., Lefsky, M., Mitchard, E. T. A., Salas, W., Zutta, B. R., Buermann, W., Lewis, S. L., Hagen, S., Petrova, S., White, L., Silman, M., and Morel, A.: Benchmark map of forest carbon stocks in tropical regions across three continents, P. Natl. Acad. Sci. USA, 108, 9899–9904, 2011.

**Influence of climate variability, fire and phosphorus limitation on the vegetation structure and dynamics in the Amazon-Cerrado border**

Emily Ane Dionizio da Silva[1], Marcos Heil Costa[1], Andrea Almeida Castanho[2], Gabrielle Ferreira Pires[1], Beatriz Schwantes Marimon[3], Ben Hur Marimon-Junior[3], Eddie Lenza[3], Fernando Martins Pimenta[1], Xiaojuan Yang[4], Atul K. Jain[5]

[1] Department of Agricultural Engineering, Federal University of Viçosa (UFV), Viçosa, MG, Brazil

[2] The Woods Hole Research Center, 149 Woods Hole Rd., Falmouth, MA USA

[3] Federal University of Mato Grosso, Nova Xavantina Campus, Nova Xavantina, MT, Brazil

[4] Oak Ridge National Laboratoty, Oak Ridge, TN, USA

[5] Department of Atmospheric Sciences, University of Illinois at Urbana-Champaign

Correspondence to: Emily Ane D. da Silva (emilyy.ane@gmail.com)

**Abstract**

[revised manuscript text omitted]

In this paper we use the dynamic vegetation model INLAND (Integrated Model of Land Surface Processes) to evaluate the influence of inter-annual climate variability, fire occurrence and P limitation in the Amazon-Cerrado transitional vegetation dynamics and structure. We assess how each element affects the net primary production (NPP), leaf area index (LAI) and aboveground biomass (AGB) and compare the model simulated AGB to observed AGB data. The results presented here are important to build models that

accurately represent the transition vegetation, and show the need to include the spatial variability of eco-physiological parameters in these areas.

**2    Materials and methods**

**2.1    Study Area**

The present study focuses on the Amazon-Cerrado transition (Figure 1). We use the official delimitation of the Brazilian biomes proposed by IBGE (2004), and define five transects along the transition border. Transects 1 to 4 are established considering approximately 330 km into the Amazon and 330 km into the Cerrado domain, while Transect 5 is 880 km long on the southern Amazon-Cerrado border. The transects are located as follows: Transect 1 (T1, 43°- 49°W; 5°- 7°S), Transect 2 (T2, 46°-51° W; 7°-9S), Transect 3 (T3, 48°-54° W; 9°-11° S), Transect 4 (T4, 49° - 55° W; 11°-13° S), and Transect 5 (T5, 53° - 61° W; 13°-15° S) (Figure 1).

**2.2    Description of the INLAND Surface Model**

The Integrated Model of Land Surface Processes (INLAND) is the land-surface component of the Brazilian Earth System Model (BESM). INLAND is based on the IBIS model (Integrated Biosphere Simulator, Foley et al., 1996; Kucharik et al., 2000), which considers changes in the composition and structure of vegetation in response to the environment and incorporates important aspects of biosphere-atmosphere interactions. The model simulates the exchanges of energy, water, carbon and momentum between soil-vegetation-atmosphere. These processes are organized in a hierarchical framework and operate at different time steps, ranging from 60 minutes to 1 year, coupling ecological, biophysical and physiological processes. The vegetation structure is represented by two layers: upper (arboreal PFTs) and lower (no

arboreal PFTs, shrubs and grasses) canopies, and the composition is represented by 12 plant functional types (PFTs) (e.g., tropical broadleaf evergreen trees or C4 grasses, among several others).The photosynthesis and respiration processes are simulated in a mechanistic manner using the Ball-Berry-Farquhar model (details in Foley et al., 1996). The vegetation phenology module simulates the processes such as budding and senescence based on empirically-based temperature thresholds for each PFT. The dynamic vegetation module computes the following variables yearly for each PFT: gross and net primary productivity (GPP and NPP), changes in AGB pools, simple mortality disturbance processes and resultant LAI, thus allowing vegetation type and cover to change with time. The partitioning of the NPP for each PFT resolves carbon in three AGB pools: leaves, stems and fine roots. The LAI of each PFT is obtained by simply dividing leaf carbon by specific leaf area, which in INLAND is considered fixed (one value) for each PFT.

INLAND has eight soil layers to simulate the diurnal and seasonal variations of heat and moisture. Each layer is described in terms of soil temperature, volumetric water content and ice content (Foley et al., 1996; Thompson and Pollard, 1995). Furthermore, all of these processes are influenced by soil texture and amount of organic matter within the soil profile.

Considering these aspects of vegetation dynamics and soil physical properties the model can simulate plant competition for light and water between trees, shrubs and grasses through shading and differences in water uptake (Foley et al., 1996). These PFTs can coexist within a grid cell and their annual LAI values indicate the dominant vegetation type within a grid cell. For example, the dominant vegetation type is a Tropical Evergreen Forest if the PFT tropical broadleaf evergreen tree has an annual mean upper canopy LAI ($LAI_{upper}$) above 2.5 $m^2$ $m^{-2}$. On the other hand, the dominant vegetation type is a Tropical Deciduous Forest if the tropical broadleaf drought-deciduous tree has an annual mean $LAI_{upper}$ above 2.5 $m^2$ $m^{-2}$. Where total

tree LAI (LAI$_{upper}$) is between 0.8 and 2.5 m$^2$ m$^{-2}$, dominant vegetation type is savanna, and LAI$_{upper}$ values smaller than 0.8 m$^2$ m$^{-2}$ characterize a grassland vegetation type.

We assume that the vegetation types Tropical Evergreen Forest and the Tropical Deciduous Forest in INLAND represent the Amazon rainforest, while Savanna and Grasslands represent the Cerrado. Savanna would be equivalent to the Cerrado physiognomies *Cerradão* and *Cerrado sensu strictu*, while Grasslands would be equivalent to the physiognomies *Campo sujo* and *Campo Limpo* (*sensu* Ribeiro and Walter, 2008).

The soil chemical properties are represented by the carbon, nitrogen and phosphorus. The carbon cycle is simulated through vegetation, litter and soil organic matter, where the biogeochemical module is similar to the CENTURY model (Parton et al., 1993; Verberne et al., 1990). The amount of C existing in the first meter of soil is divided into different compartments characterized by their residence time, which can vary in an interval of hours for microbial AGB and organic matter to several years for lignin. The model considers only the soil N transformations and carbon decomposition, but the N cycle is not fully simulated and N does not influence the vegetation productivity, i.e., there is a fixed C:N ratio. P is used only to limit the gross primary productivity. The total P available in the soil (P$_{total}$) is used to estimate the maximum capacity of carboxylation by the Rubisco enzyme (V$_{max}$) through a linear relationship.

$$V_{max} = 0.1013\,P_{total} + 30.037 \tag{1}$$

where V$_{max}$ and P$_{total}$ are given in μmolCO$_2$ m$^{-2}$ s$^{-1}$ and mg kg$^{-1}$, respectively. This equation has been developed by Castanho et al. (2013) based on data for tropical evergreen and deciduous trees, and is applied only to these two PFTs in the model.

INLAND also contains a fire module, from the Canadian Terrestrial Ecosystem Model CTEM (Arora and Boer, 2005). In this module, three aspects of the fire triangle are considered – the availability of fuel to

burn, the flammability of vegetation depending on environmental conditions, and the presence of an ignition source. The natural ignition probability is summed to arbitrary anthropogenic fire probability, and the burned area is modeled as an ellipse of dimensions determined by wind and fuel conditions (Arora and Boer, 2005).

**2.3 Observed data**

**2.3.1 Phosphorus databases**

We used two P databases to estimate $V_{max}$ (Equation 1): one regional (referred to as PR) and one global database referred to as PG). In addition, a control P map (PC) represents the unlimited nutrient availability case, equivalent to a $V_{max}$ of 65 $\mu molCO_2$ m$^{-2}$ s$^{-1}$, or 350 mg P kg$^{-1}$ soil, according to Equation 1.

The PR database was developed from total P in the soil for the Amazon basin published by Quesada et al. (2011) plus 54 additional available P samples (P extracted via Mehlich-1 extractor, P $_{mehlich-1}$) (Figure 2a). We used the P$_{-mehlich-1}$ and clay contents measured in a forest-savanna transition region in Brazil (Mato Grosso state) to estimate $P_{total}$ and expand the coverage area of the P data (Section S1). These 54 samples were gridded to a $1° \times 1°$ grid to be compatible with the spatial resolution used by INLAND, resulting in 12 additional pixels with observed total P content (Figure 2a). For pixels without observed $P_{total}$, the $P_{total}$ was assumed to be 350 mg P kg$^{-1}$ soil, similarly to the PC conditions.

A global dataset of $P_{total}$ (Figure 2b) was also used to estimate $V_{max}$. This global data set is part of a database containing six global maps of the different forms of P in the soil (Yang et al., 2013). The $P_{total}$ was estimated from lithologic maps, distribution of soil development stages, fraction of the remaining source material for different stages of weathering using chronosequence studies (29 studies), and P distribution in different forms for each soil type based on the analysis of Hedley fractionation (Yang and Post, 2011), which are part of a worldwide collection of soil profile data. The uncertainties and limitations associated with this

database are restricted to the Hedley fractionation data used, which are 17% for low weathered soils, 65% for intermediate soils and 68% for highly weathered soils (Yang et al., 2013).

**2.3.2 Above-Ground AGB (AGB) database**

The AGB database used was created by Nogueira et al. (2015) and considered undisturbed (pre-deforestation) vegetation existing in the Brazilian Amazonia. This database was compiled from a vegetation map at a scale of 1:250000 (IBGE, 1992) and AGB averages from 41 published studies that had conducted direct sampling in either forest (2317 plots) or non-forest or contact zones (1830 plots). We bi-linearly interpolated the AGB (dry weight) for each transect considering $1° \times 1°$ to ensure compatibility of the observed and simulated data.

Five longitudinal transects (Figure 1) were used separately to characterize AGB in the Amazon-Cerrado border (Figures 3a and 3b). In T1, T2, T3 and T4, the higher AGB values in the west and lower values in the east are consistent with the transition from a dense and woody vegetation (the Amazon forest) towards a sparse vegetation with lower AGB (the Cerrado). However, T1 shows a more gradual reduction of AGB along the west to east gradient, while in T2, T3 and T4 where the transition is more abrupt. In T5 no west-east gradient is present with high AGB heterogeneity and predominant low AGB across the transect (Figure 3b).

**2.4 Simulations**

The model was forced with the prescribed climate data based on the Climate Research Unit (CRU) database (Harris et al., 2014). Two climate boundary conditions were used: the first is referred to as the

monthly climatological average (CA) that represents the average climate for the period 1961-1990. The second climate boundary condition is the historical dataset, for the continuous period between 1948 and 2008 (CV). For both boundary conditions, the variables used are rainfall, solar radiation, wind velocity and maximum and minimum temperatures. The CRU database has been widely used by the scientific community in case studies, because these data preserve the spatial mean of the rainfall data, although, they do not provide adequate representation of their variance precipitation (Beguería et al., 2016). The dataset has a 1-degree spatial resolution and a monthly time resolution.

Soil texture data is based on the IGBP-DIS global soil (Global Soil Data Task 2000) (Hansen and Reed, 2000). The model simulations were run for the time period 1582-2008, a total of 427 years. In the CV group of runs, the model was spin-up by cycling the 1948-2008 climate data (61-year) seven times, totaling 427 years. In the CA group of runs, the annual mean climate data was cycled 427 times. In both cases, $CO_2$ varied from 278 to 380 ppmv, according to observations in the period, updated annually. In both cases, only the model results of the last 10 years were used to analyze the results.

The experiment design is a factorial combination of the climate scenarios (CA, monthly climatological average, 1961-1990; CV, monthly climate time series, 1948-2008), the nutrient limitation on $V_{max}$ (PC, no P limitation ($V_{max} = 65$ $\mu molCO_2$ m$^{-2}$ s$^{-1}$); PR, regional P limitation; PG, global P limitation) and the occurrence of fire (F) or not (Table 1). The 12 combinations in Table 1 allow the evaluation of individual and combined effects of climate, soil chemistry, and the incidence of fire on the variables: Net Primary Production (NPP), tree AGB, and LAI of the upper and lower canopies (LAI$_{upper}$, LAI$_{lower}$).

We consider that the subtraction between the simulations $(CV+PC) - (CA+PC) = (CV-CA)|_{PC}$ represents the isolated effect of inter-annual climate variability without P limitations. The same logic is

applied to isolate other factors such as fire and P in different climate scenarios. For example, the fire effect under average climate without P limitation case is calculated by the difference between CA+PC+F and CA+PC, so that (CA+PC+F)–(CA+PC) = $F|_{CA,PC}$. Similarly, the isolated effect of fire under a climate with inter-annual variability scenario without influence of P limitation is calculated by the difference between CV+PC+F and CV+PC, so that (CV+PC+F)–(CV+PC) = $F|_{CV, PC}$. The different combinations of climate scenarios with and without fire effects and with and without P limitations are described in Table 2.

**2.5    Statistical analysis and determination of the best model configuration**

The statistical analysis is divided in four parts. First, we present maps of the isolated effects for all simulated area calculated as the average of last ten years of simulated spatial patterns. The statistical significance of the isolated effects on NPP, LAI and AGB are determined using the t-test with $p < 0.05$. The results are tested in each pixel, for all the simulated domain (n = 10).

Second, we present an analysis of variance using the one-way ANOVA and the Tukey-Kramer test in the transition zone. We consider all 31 pixels which fall in transects T1 to T5 ($n_{pixels}$). The results presented are based on the set of last 10 years of simulation (1999-2008, $n_{years}$) for the 12 combinations ($n_{simulation}$) in Table 1. Moreover, we grouped treatments according to climate regardless of P limitation, presence or absence of fire, where all sets with CV vs CA are tested (Group 1, n=1860, ($n_{pixel}$ x $n_{year}$ x ($n_{simulation}$/2))). Similarly, in Group 2 we tested if PC, PR or PG were significantly different from each other regardless the F or climate used (Group 2, n=1240, ($n_{pixel}$ x $n_{year}$ x ($n_{simulation}$/3))). In Group 3 we tested if fire introduced a significant effect regardless of climate and P limitation (Group 3, n=1860, ($n_{pixel}$ x $n_{year}$ x ($n_{simulation}$/2))). Finally, all treatments were tested to each simulation assessing their individual effects on NPP, LAI and AGB ($n_{pixel}$ x $n_{year}$ = 310).

Third, a correlation coefficient between the simulated and observed values for AGB was calculated for each transect. The simulated variables are averaged for the last 10 years of simulations (1999 - 2008) and compared to AGB from Nogueira et al. (2015) within a grid cell.

Finally, we evaluate INLAND's ability to assign the dominant vegetation type by analyzing 10 years of probability of occurrence. If the dominant vegetation type (evergreen tropical forest, or deciduous forest for the Amazon rainforest, and savanna or grasslands for Cerrado) in a pixel is the same in more than 90% of the simulated years (9 out of 10), then the simulated vegetation type is defined as "very robust" for that pixel; if it occurs in 70 – 90% of the simulated years, the simulated result is considered to be "robust". If the dominant vegetation occurred in less than 70% of simulated years, the pixel is considered "transitional" vegetation.

**3    Results**

**3.1    Influence of climate, fire and phosphorus in the Amazon-Cerrado transition region**

**3.1.1    Spatial patterns**

Overall, the inclusion of inter-annual climate variability (CV) resulted in a decrease in the simulated average tree biomass (TB) by 3.8% in Amazonia, and by 8.7% in Cerrado in comparison to average climate (CA) (Figure 4a). The spatial differences between CV and CA for TB simulations are statistically significant and range from -3 kg-C m$^{-2}$ to +2 kg-C m$^{-2}$. The state of Pará, with higher influence of the El Niño phenomenon, experienced the highest decrease in TB in the CV simulation. In the state of Roraima, on the other hand, there was an increase of about 2 kg-C m$^{-2}$ in TB when CV was considered. Bolivia and southwest

of Mato Grosso state also presented, in some grids points, a significant increase in AGB higher than 2 kg-C m$^{-2}$.

On average, P acts as a limiting factor in the simulated TB, decreasing by 13% in regional P (PR) simulation and 15% in global P (PG) simulation. In PR, TB decreased mainly in the southeastern Amazonia (between Pará and northeastern Mato Grosso states) and northwestern Amazonas state (Figure 4b). In PG, the largest TB decline occurred in central Amazonia, northeastern Pará and northeastern Mato Grosso (Figure 4c). In Cerrado, on the other hand, TB declined by 2% for PR and 9% for PG with respect to the control simulation. In PR, the few pixels in the Cerrado that have P limitation showed a significant decrease in TB (Figure 4b), while in PG the TB reduction was statistically significant for most of the Cerrado domain, except in southern Tocantins state (Figure 4c).

The tree biomass reduction due to fire events is much higher in magnitude more than due to P limitation or inter-annual climate variability (Figure 4d). The greater water availability is related to small or null fire effect in the Central Amazon rainforest agrees with the fact that Amazonia is naturally inflammable as well as a gradient towards seasonally dryer climate that increases the intensity and magnitude of fire effects towards the Cerrado (Figure 4d). The fire effect on TB over the Amazon domain was 21-24% of the P limitation effect (range for PR and PG cases), while the fire effect on TB over the Cerrado was more than 250% of the P limitation effects in CV simulations, which is due to quick growth of grasses after fire occurrence in the latter.

**3.1.2   Influence of climate, fire and phosphorus in the transects**

Results of the ANOVAs and Tukey-Kramer test indicate that the inclusion of CV, limitation by P (PR and PG) and fire in INLAND led to significantly different averages of NPP, LAI and AGB in the transition

zones. This influence of climate, P and fire are shown separately in Tables 3 to Table 5 and combined in Table 6.

The effects of climate and P on productivity show that CV reduces the NPP from 0.68 kg-C m$^{-2}$ yr$^{-1}$ to 0.64 kg-C m$^{-2}$ yr$^{-1}$ (Table 3) and the P effect results in NPP decline from 0.71 kg-C m$^{-2}$ yr$^{-1}$ to 0.64 kg-C m$^{-2}$ yr$^{-1}$ (both PR and PG) (Table 4). The fire effect, moreover, has a positive effect on NPP from 0.66 kg-C m$^{-2}$ yr$^{-1}$ when fire is off to 0.67 kg-C m$^{-2}$ yr$^{-1}$ when fires is on. This difference, albeit low, is statistically significant (Table 5).

In addition CV and P limitation reduce the LAI$_{total}$ in the canopy (Table 3 and Table 4), increasing three times LAI$_{lower}$ and decreasing LAI$_{upper}$ (Table 5). The magnitude of fire effect on AGB (46.7%, Table 5) is greater in relation to the CV (5%, Table 3) and P (14%, Table 4) limitation effects.

Even though CV effects on NPP and AGB for each simulation is not statistically significant, the effects of fire and P limitation (regardless of phosphorus map) are. Fire effects are significant only for structural variables as AGB, LAI$_{total}$, LAI$_{upper}$ and LAI$_{lower}$. It presents an increase of LAI total of 1.52 m$^2$ m$^{-2}$ in CV+PG+F in relation to CV+PG, and of 1.32 m$^2$ m$^{-2}$ in CV+PR+F in relation to CV+PR (Table 6).

**3.1.3 West-East patterns of AGB in the Amazon-Cerrado transition**

The model used in this study simulates > 80% of the observed AGB variability in all treatments along the transition area except in T5 (Table 7). It shows that the model is able to capture AGB variability along the transition area, which is relevant when compared to studies that simulate 50% of the observed AGB variability (Senna et al., 2009; Castanho et al., 2013).

It is not possible to identify a treatment that best represents AGB in all transects (Table 7). A combined analysis of Table 7 and Figure 5 indicates a general agreement that observed AGB decreases from W to E in

T1 to T4, and this is well captured by several configurations of the model, with specific differences among them. Overall, CA and PC configurations, being the least disturbed treatments, yield higher AGB, while the introduction of CV, PG and F reduce the AGB. However, the simulated results may be above or below the observed ones, which suggests that additional local factors are not included in the model.

The curves of AGB (Figure 5) show the impact of CV, PG and F along the W-E transition. PG has a high influence on the transition, decreasing the ABG especially in the western part of the transects, where the Amazon vegetation is predominant. This feature is particularly simulated in T3 and T4, where PG decrease the AGB by 2 kg-C m$^{-2}$ in the west pixels of these transects (Figure 5). In T1, T2 and T5, AGB decline is also higher with P limitation when compared to the curves limited only by CV. However, in T1 model simulations tend to underestimate the highest and the lowest AGB extremes, and the absolute values were always underestimated, despite the improvement in correlation with the inclusion of the fire component (Table 7).

Fire, however in T2, T3 and T4, is responsible to approach the simulated AGB to the observed AGB in the eastern pixels into Cerrado domain (Figure 5). In T5 these relations are similar, with climate presenting less influence on AGB decrease than P, and fire appears mainly as a reducer factor.

**3.2 Simulated composition of vegetation**

Most of the pixels in CA show very robust simulations, with more than 90% of the same vegetation cover in the simulated last 10 years (Figure 6a-c and 6g-i). A larger number of pixels with transitional vegetation were simulated in CV (Figure 6d-f and and 6j-l). An even higher variability in CV compared to CA simulations was observed when we added the effects of P limitation and fire (Figure 6a and 6j-l).

The vegetation composition in all P limitation scenarios for CA simulations resulted in robust simulations for nearly all pixels, except for the north of Cerrado domain (Figures 6a, 6b and 6c). The CA+PC and CA+PR simulations had the same vegetation composition, while CA+PG replaced the deciduous forest by evergreen forest in the central Cerrado region, around 8°S 46°W (Figures 6A, 6B and 6C). This behavior might be related to the higher $P_{total}$ values in PG than PR and PC for the Cerrado region (Figure S1). Cerrado was better represented in CV+PC, CV+PR and CV+PG than in the same CA combinations (Figure 6). The occurrence of forested areas in central Cerrado decreased in CV combinations, these being replaced by the savanna or grassland vegetation class.

When the effect of fire was added to CA simulations, the model simulated an increase in the uncertainty on the vegetation cover classification in the Cerrado region. The effect of fire reduced the presence of deciduous forest in central Cerrado biome as well as in CA+PC, and the vegetation was replaced by evergreen forest and savanna in CA+PC+F (Figures 6G, 6H, 6I). In CV simulations, fire effect results in the replacement of the deciduous and perennial forest by savanna and grasses in all central Cerrado region (Figures 6J, 6K and 6L).

For all combinations used, transitional forest areas in the northern and southwestern Cerrado biome are not adequately represented. With >90% of concordance, INLAND assigns the existence of tropical evergreen forest rather than deciduous forest in some pixels in the north of the transition, and the existence of tropical evergreen forest rather than savanna in the southwest, indicating difficulty to simulate transitional vegetation in these regions.

**4  Discussion**

The inclusion of CV, PR and PG and fire in INLAND showed significant influences on the simulated vegetation structure and dynamics in the Amazon-Cerrado border (Figure 4 and Table 6), suggesting that these factors play key role on vegetation structure in the forest-savanna border and can improve the simulated representation of the current contact zone between these biomes. This is broadly consistent with the literature that investigated causes of savanna existence in the real world (Hoffmann et al., 2012; Dantas et al., 2013; Lehmann et al., 2014). In this study, the spatial analysis and the Tukey-Kramer test (TK) show a difference in magnitude among these factors in vegetation, with fire occurrence and P limitation being stronger than inter-annual climate variability along the transects (Figure 4).

The spatial analysis showed that CV declines AGB predominantly in eastern Amazonia (Figure 4a). Climate of this region is intensely affected by El Niño–Southern Oscillation (ENSO), which could reduce precipitation by 50%, placing the vegetation under intense water stress (Botta and Foley, 2002; Foley et al. 2002; Marengo et al., 2004; Andreoli et al., 2013; Hilker et al., 2014). This reduction in rainfall in dry years brings in direct changes in carbon flux (NPP) and stocks in leaves and wood, leading to changes in vegetation structure. In addition to inter-annual changes in the rainfall, inter-annual variability in other climate variables in CV also affect AGB, as average, maximum and minimum temperature, as well as wind speed and specific humidity, and influence photosynthesis on the model both directly (through Collatz and Farquhar equations) and indirectly (e.g. through evapotranspiration). Our results showed significant differences for most part of the biomes, except central Amazonia (Figure 4a), where CV and precipitation seasonality have been pointed as secondary effects on vegetation (Restrepo-Coupe et al., 2013), since there is no shortage of water availability during the dry season.

Along the Cerrado, lower water availability in some years in CV affects tree biomass, although that vegetation is predominantly grassy-herbaceous. The AGB decline is significant for most part of the simulated Cerrado domain (Figure 4a) and average values could represent half the amount of typical tree biomass in this biome. This reduction in AGB reflects INLAND's ability to simulate similar Cerrado conditions and expose the few trees to high water stress.

Throughout the transects, however, no significant difference was found for average AGB between CV+PC and CA+PC by TK at $p<0.05$ (Table 6). On the other hand, when we analyzed the influence of CV for the same pixels, but using all simulations (Table 3), regardless of P limitation and fire occurrences, the results showed that the decrease in AGB by 0.38 kg-C m$^{-2}$ (5.7%) is statistically significant along the transition.

P limitation effect was statistically significant for PR and PG along all the Amazon domain and the main differences between these simulations were the spatial patterns of tree AGB decrease (Figure 4b and Figure 4c). We cannot affirm which of these databases is better because they are the results of different methodologies and observations (Quesada et al., 2009; Yang et al., 2014). However, PG showed a higher AGB decrease in central Amazonia, northeastern Pará and northeastern Mato Grosso state, indicating that in these areas the P limitation is higher. This result does not corroborate the northwest-southeast AGB gradient found in the Amazon basin, which showed a higher productivity in the west where soils are more fertile than those found in the southeast (Aragão et al., 2009; Saatchi et al., 2007; Nunes et al., 2012; Lee et al., 2013). On the other hand, PR AGB agrees with the northwest-southeast gradient, presenting less limitation in the soils of central Amazonia with declines in AGB mainly in the southeastern part of the rainforest (between Pará and northeastern Mato Grosso states) (Figure 4b).

In Cerrado, P limitation also influenced vegetation (Figure 4c) and presented statistically significant differences when compared to CV+PC. In this biome, as well as in the Amazon, tree abundance richness and diversity have been generally associated with increases of soil fertility (Long et al., 2012; Vourtilis et al., 2013), highlighting the importance of P in the composition and maintenance of vegetation, especially in transition areas.

Compared to the Amazon domain, the magnitude of effects of P limitation is lower in the Cerrado. However, few pixels in PR that have P limitation showed a significant decrease in arboreal AGB (Figure 4b), while in PG, we found reduction of AGB for most of the Cerrado domain, except only for the southern Tocantins state (Figure 4c). Despite the differences in spatial patterns, there was no statistically significant differences between PR and PG within the transects (Table 4 and Table 6).

The spatial difference between PG and PR showed that PG is lower than PR in the western Amazonia, and higher in northern Amazonia. Moreover, PG have low P values in south of the transition compared to PR, while in Cerrado domain P values ranged between 120 to 200 mg kg$^{-1}$ (Figure S1). Although the PR dataset includes every known P data collected in the region, these differences reinforce the need to improve the data of $P_{total}$ in the soils of the Amazon and Cerrado/Amazon transition domains. Currently, $P_{total}$ data in Cerrado is scarce, and make unfeasible to establish a proxy similar to Castanho et al. (2013), which was specific for the Amazon.

To our knowledge, the most part of the Dynamic Global Vegetation Models (DGVMs) do not consider the complete phosphorus cycle, despite the importance of nutrient cycling for AGB maintenance and tropical vegetation dynamics in dystrophic soils. For example, nutrient cycling in the Amazon/Cerrado transition is closely related to the hyper-dynamic turnover of the AGB (Valadão et al. 2016), in which some key species

might also be crucial to the hyper-cycling of nutrients through which vegetation sustain the constant input of nutrients, including large annual amounts of available P (Oliveira et al. *in press*).

The decrease in tree AGB occasioned by P limitation can contribute to a decrease in litter production and consequently could affect nutrient cycling in tropical ecosystems. According to Oliveira et al. (*in press*), the litter produced by vegetation corresponds to the main return route to the available fraction of P for plants, especially in the transition areas, where $P_{available}$ in the soil is very low. In our model, however, P acts directly in the photosynthesis limitation through $V_{max}$ and cannot be reabsorbed by the roots. Thus, the litter produced in vegetation contribute only to dry matter and fire occurrence increase. In nature, the litter affected by fire occurrence volatilizes the small amount of P available to plants, increasing the nutrient losses of the ecosystem. Despite this simplified representation in INLAND, it can represent the P influence on woody AGB in the Amazon and Cerrado.

The fire occurrence is an important factor controlling the AGB dynamics in the Cerrado or in the transition vegetation (Silvério et al., 2013; Couto-Santos et al., 2014; Balch et al., 2015), which this study clearly replicates, showing statistically significant influences when compared to control simulations (Figure 4d and Table 5). In the transition, the fire effect may reduce average AGB by 50%, which under climate change or deforestation conditions may lead to an even stronger change in the vegetation structure and dynamic. In the Cerrado domain, the simulated fire effect implies in significant increases of shrubs and herbaceous vegetation and decreases of the arboreal component. In nature, however, the Cerrado is relatively resilient to fire depending on the velocity, intensity and duration of the burning (Rezende et al., 2005; Elias et al., 2013 Reis et al., 2015). The adaptive morphological nature and the low nutrient requirement of vegetation allow Cerrado the capacity to rapidly restore the vegetation after fire occurrence (Hoffmann et al.,

2005; Hoffmann et al., 2012). In our model, the restoration of vegetation after fire occurrence is exclusively due to the canopy opening and consequently more luminosity penetration into lower canopy.

This study shows an improvement in the correlations between simulated and observed AGB when compared to previous modeling studies, regardless of treatment, with correlation coefficients usually above 0.80 for the transects, except for T5, for which the correlation coefficient value is usually below 0.5 (Table 7). Senna et al. (2009) found 0.20 as maximum correlation coefficient between simulated and observed ABG while Castanho et al. (2013) showed 0.80 for Amazonia domain. From Figure 5, it is clear that CV, F and P limitation in the transition zone reduce the AGB, approaching the simulated to the observed data, and play important roles in the simulations, but the only inclusion of these effects is still insufficient to represent the actual vegetation structure in the Amazon-Cerrado border (Figure 6L). In our interpretation, this means that other important factors are still missing from the simulation, especially in T5, where soils are rocky and shallow. A better spatial representation of soil physical properties, including shallow rocky soils, as well as spatially varying physiological parameterizations of the vegetation such as carbon allocation, deciduousness of vegetation, and residence time are probably needed to improve the simulations, in particular in the northern and southern extremes of the border (T1 and T5).

In addition, literature shows that in the transition area, soils are very different than Amazon soils, and that essential proprieties for modeling are peculiar (Silva et al., 2006; Vourlitis et al., 2013; Dias et al. 2015). For example, Dias et al. (2015) recently showed that the pedological functions normally used by DGVMs may underestimate the saturated hydraulic conductivity ($K_s$) by >99%, transforming a well-drained soil with $K_s = 1.5 \cdot 10^{-4}$ m.s$^{-1}$ (540 mm.h$^{-1}$) in reality into an impervious brick with $K_s = 3.3 \cdot 10^{-7}$ m.s$^{-1}$ (1.2 mm.h$^{-1}$) in the model.

For all transects, the AGB curves have similar patterns (Figure 5); the smaller difference is observed between CA+PC and CV+PG curves, while the larger difference is when fire is present. The effect of P limitation appears as an effect of intermediate magnitude, reducing the AGB by more than the effect of inter-annual climate variability. In the east, it is observed that there is little or no difference among AGB simulated by CA+PC, CV+PC and CV+PG, revealing that inter-annual climate variability and P have smaller influence in the AGB. However, in the east of T2, T3 and T4, fire is the factor that adjusts the simulated to the observed data (Figure 5), differently than the grid points in the West, where CV+PG is a better proxy between observed and simulated data.

Such conditions are interesting because they reflect the different mechanisms that regulate the structure of these ecosystems and probably the phytophysiognomies distribution. For example, P limitation seems to be the factor that improves simulated AGB in regions where the predominant vegetation type is the tropical rainforest. Fire, on the other hand, improves the AGB in grid points where the Cerrado occurs. Moreover, important factors such as productivity partitioning into leaves, roots and wood carbon pools are assumed to be fixed in space and time within a given PFT, neglecting the natural capacity of transitional forests to adapt itself and to adjust their metabolism to local environmental conditions (Senna et al., 2009). In years of severe drought, transitional forests could prioritize the stock of carbon to fine roots instead of the basal increment to maximize access to available water, or make hydraulic redistribution to maintain the greenness and photosynthesis rates. Brando et al. (2008) found high sensitivity in carbon allocation for eastern Amazon basin trees, which reduced wood production by 13-60% in response to an artificial drought. Although in INLAND soil moisture can reduce the photosynthetic rates during the months of lower rainfall,

it does not dynamically change the allocation rates, exposing the PFTs in these areas to severe water stress and underestimating the AGB, such as in the west of T1 (Figure 5a).

T2, T3 and T4, located in the central part of the Amazon-Cerrado transition, showed the highest average correlations between observed and simulated data (Table 7). For these transects, INLAND seems to be able to capture the high variability of AGB gradient.

At T5, located at the south of the transition, the average correlations were low for all treatments, indicating that INLAND has difficulty to represent the AGB gradient there (Table 7). However, it captures the lower AGB as compared to the northern ones. In this region, the vegetation is characterized by a wide diversity of physiognomies, which varies with other preponderant factors, such as lithology, soil depth, topography and fertility. The observed data also showed high AGB variability, indicating that there are changes in the vegetation structure, featuring medium-sized and small vegetation types on different soil types. In INLAND, however, features such as lithology and water-table depth are not considered due to the complexity of its representation on the large scale, limiting the representation of a heterogeneous environment throughout the transition.

Different patterns of vegetation distribution along the Amazon-Cerrado border exist and are influenced not only by inter-annual climate variability, P limitation, and fire, but also by the ecophysiological parameters, which may have different behavior according to the environmental conditions and soil proprieties. Obtaining these parameters is a challenge to the scientific community once the field measurements are difficult due to the extension of the transition area. More observed data are needed to establish and implement the plasticity of the fixed parameters such as carbon coefficients allocation, residence time, dependence of the deciduousness on P, among others.

Another point to discuss is that the model simulates, in a few pixels in southeastern Cerrado, very robust simulations of the presence of savanna and grassland even in the absence of fire (Figure 6A-F and 6a-f). This is, in our view, a result of the intense water and heat stress in this region. In the Brazilian Cerrado, the high temperatures (> 35 °C) combined to the dry season duration (as long as 6 months with little or no rain) exposes the vegetation to a severely stressed situation, so that a low biomass, low LAI vegetation may exist without the need of a frequent disturbance.

**5    Conclusions**

This is the first study that uses modeling to assess the influence of inter-annual climate variability, fire occurrence and phosphorus limitation to represent the Amazon-Cerrado border. This study shows that, although the model forced by a climatological database is able to simulate basic characteristics of the Amazon-Cerrado transition, the addition of factors such as inter-annual climate variability, phosphorus limitation and fire gradually improves simulated vegetation types. These effects are not homogeneous along the latitudinal/longitudinal gradient, which makes the adequate simulation of biomass challenging in some places along the transition. Our work shows that fire is in the main determinant factor of the vegetation changes along the transition. The nutrient limitation is second in magnitude, stronger than the effect of inter-annual climate variability.

[revised manuscript text omitted]

---

## Author Comment (AC2) · 24 Mar 2017

Response to Reviewer on Paper doi:10.5194/bg-2016-510

(Reviewer comments in *italics;* Responses in **bold**)

Response to Anonymous Referee #2

**We are grateful to the reviewer for your insightful comments and helpful suggestions. Please find detailed responses to each comment below.**

*General comments*
*This study analyses the INLAND vegetation model with the purpose of discerning the relative impacts of fire, empirical phosphorus limitation and climate variability on predictions of ecosystem structure across forest-cerrado transitions in S. America.*
*In common with reviewer #1, I think that the text requires careful editing, particularly for (mostly minor but widespread) grammatical errors.*

**Response: We apologize for the mistakes. The manuscript has been completely rewritten. We hope it will satisfy the reviewer now.**

*The model description is extremely vague, and parameterization and calibration carried out prior to these experiments is omitted. I am skeptical that the model simply performed reasonably the first time that it was run.*

**Response: Throughout the responses to the reviewer specific comments, we have provided several reasons for a non-detailed description of the INLAND model, and incorporated these explanations into the revised manuscript.**

*What uncertainties do you need to grapple with before the model output falls within the sensible range? Without this information, the reader might assume that goodness-of-fit tests between the models and the observations might have been substantially affected by undisclosed model calibration. Given that, I find the comparison of different influences over model outputs (fire, phosphorus, etc.) to be somewhat predictable and not very interesting.*

**Response: We are using the same calibration used by Senna et al. (2009). We have not calibrated the model specifically for this run, as the reviewer implies.**

*The discussion section contains numerous logical errors confusing the output of the model and the behaviour of the ecosystem in real life. Until these are rectified, I do not think that this paper is of sufficiently high scientific standard to be published.*

**Response: The manuscript has been completely rewritten. We hope it will satisfy the reviewer now.**

*The reliance on statistical tests is distracting. A better analysis of the consequences of and the uncertainty in the impacts would be much more useful.*
**Response: We are not sure we understand this comment. Reliance on statistical tests is a standard in science.**

*Specific Comments.*

*L112: Is Kucharik (2000) really the most recent reference for the INLAND model? I'm fairly sure this isn't the case. To be repeatable, this model description needs to provide at a minimum references to the most recent version, along with specific descriptions of the model equations and parameters if they have been modified since the last publication. Many EGU journals stipulate that directions to the code used are also included. I do not know if this applies to BGD, but it would be good practice to do so.*

**Response: The INLAND project was mainly a revision of the IBIS code, through assembly and standardization of different IBIS versions, and improvements in software engineering. We used the version described by Senna et al. (2009) as starting point for INLAND. No changes in tuning were done since that paper. Some of the key equations and parameterizations, however, were described by Kucharik et al. (2000). The code can be downloaded from http://biosfera.dea.ufv.br/en-US, clicking on models and then on INLAND.**

*L115: Does this mean the vegetation types compete for light, or for water & nutrients?*
*The mechanisms of competition and dynamic vegetation are a critical part of a model of this type. I am surprised you skipped over this so briefly.*

**Response: To clarify this, the revised manuscript will include additional explanations, which can be found below:**

   **"The vegetation structure is represented by two layers: upper (arboreal PFTs) and lower (no arboreal PFTs, shrubs and grasses) canopies, and the composition is represented by 12 plant functional types (PFTs) (e.g., tropical broadleaf evergreen trees or C4 grasses, among several others).The photosynthesis and respiration processes are simulated in a mechanistic manner using the Ball-Berry-Farquhar model (details in Foley et al., 1996). The vegetation phenology module simulates the processes such as budding and senescence based on empirically-based temperature thresholds for each PFT. The dynamic vegetation module computes the following variables yearly for each PFT: gross and net primary productivity (GPP and NPP), changes in AGB pools, simple mortality disturbance processes and resultant LAI, thus allowing vegetation type and cover to change with time. The partitioning of the NPP for each PFT resolves carbon in three AGB pools: leaves, stems and fine roots. The LAI of each PFT is obtained by simply dividing leaf carbon by specific leaf area, which in INLAND is considered fixed (one value) for each PFT.**

INLAND has eight soil layers to simulate the diurnal and seasonal variations of heat and moisture. Each layer is described in terms of soil temperature, volumetric water content and ice content (Foley et al., 1996; Thompson and Pollard, 1995). Furthermore, all of these processes are influenced by soil texture and amount of organic matter within the soil profile.

Considering these aspects of vegetation dynamics and soil physical properties the model can simulate plant competition for light and water between trees, shrubs and grasses through shading and differences in water uptake (Foley et al., 1996)."

*L116: This classification seems arbitrary to me. Why not just report the LAI numbers?*
**Response: We cannot only report LAI values because different PFTs can coexist within a grid cell. In INLAND the annual tree LAI values indicate which vegetation type dominates. Thus, the vegetation type depends not only on LAI, but also on the existence of a dominant tree PFT.**

*L122: Again, I'm not sure of the need for this cross-referencing of PFTs, 'vegetation types' (why not ecosystem type - that would be less confusing) and then names for the ecosystems. The purpose of a mechanistic model is to describe the system quantitatively and in multiple dimensions. Introducing a simplistic written classification scheme does not seem. to add any extra information.*

**Response: We are indeed describing the system quantitatively, using dynamic (NPP) and structural variables (LAI, AGB). The simplistic classification aims at summarizing the model output only. Nevertheless, all the other quantitative information is available in the manuscript. Moreover, we believe that changing the model classification to "ecosystem types" would lead to a misinterpretation of the model representation, as ecosystem includes a wide community of living and not living organisms, which are not represented in INLAND, as animals and river flows for example. Therefore, "vegetation types" is the most appropriate term, as it reflects exactly the modeling scope: the vegetation only**

*L136: From where did this relationship between P and Vcmax arise? Is it sensible to use is across this biome? More detail is needed in addition to giving the reference, given how central this relationship is to the rest of the analysis.*

**Response: This relationship is widely discussed in Castanho et al., (2013), whose results based our investigation. This information is explicit in the manuscript, please check section 2.2.**
**We report here a brief summary of how this relationship has emerged:**

- **Fyllas et al. (2009) showed that soil fertility is one of the most important predictors for observed higher nutrient concentration in Amazon tree leaves.**

- **Mercado et al. (2009) noted a correlation between observed Vcmax and P concentration in Amazon tree leaves.**

- **Castanho et al., (2013) developed a similar regression equation to that of Mercado et al. (2009, 2011) but between Vcmax and total P concentration in soil instead of the P concentration in leaves (Figure. 3b, Equation. 4 in Table 1 in Castanho et al., 2013). The advantage of this empirical regression with respect to soil P is the ability to estimate Vcmax for the whole Amazon.**

*L140: Again, how it is similar to CTEM? How dos the arbitrary ignition scheme work?*
*If this is covered later in the text, it should be referenced here.*

**Response: INLAND incorporates all fire components of the CTEM (Canadian Terrestrial Ecosystem Model) model (Arora and Boer, 2005). These components simulate fire at the daily timescale (instead of the yearly timescale of earlier models) by computing the probability of fire occurrence, which is based on biomass availability, flammability and ignition source (using observed lighting frequency). Burned area is modeled as an ellipse of dimensions determined by wind and fuel conditions. The fire model of CTEM uses an arbitrary anthropogenic fire probability which is summed to the natural ignition probability. These arbitrary ignition scheme is extensively described in Arora and Boer, 2005 page 5, but we report here a brief summary:**

- **The natural ignition probability is represented by a lightning scalar, which varies from 0 to 1 as cloud-to-ground lightning frequency varies from a specified lower value of essentially no lightning to an upper value close to the maximum observed.**

- **The probability of fire ignition due to human causes may be selected depending on location and human activity and determines the lower limit of ignition constraint. In INLAND we use 0.50 for the probability of fire ignition due to human cause.**

- **The Figure 3c the Arora and Boer, 2005 resume this relationship. See below:**

[Figure]

**Source: Arora and Boer, (2005)**

*L145: Why bring up the two options if only CTEM is used in this study?*
**Response: We agree. We don´t mention the other option anymore.**

*L185-192: This description seems more like a discussion than methods. Also, can you clarify the impact of land use on these transitions?*
**Response: We agree with the reviewer. This has been changed in the revised manuscript. This manuscript does not include any form of land use, dealing only with the natural dynamics of vegetation.**

*L194-197: I'm not sure what point you are trying to make here.*
**Response: That was unclear indeed. We modified the manuscript.**

*L212: This description of the model experiments needs cleaning up. Only the P limitation scenarios seems to have label (PC, PR, etc.) and what the combinations are is not discussed at all in the text, nor are the number of scenarios, etc.*

**Response: The labels are assigned at section 2.4 and in the Table 1.**

*L220: It is not yet clear how the model distinguishes upper and lower canopy LAI?*
*Therefore this distinction is not useful yet as a diagnostic.*

**Response: The LAI$_{upper}$ and LAI$_{lower}$ are different variables in the model, which are individually and parallelly simulated by INLAND. The LAI$_{upper}$ refers to the upper canopy represented by arboreal PFTs, while the LAI$_{lower}$ is represented by no arboreal PFTs such shrubs and grasses. To clarify this, we provided changes in the revised manuscript, it can be found below:**

**"The vegetation structure is represented by two layers: upper (arboreal PFTs) and lower (no arboreal PFTs, shrubs and grasses) canopies, and the composition is represented by 12 plant functional types (PFTs) (e.g., tropical broadleaf evergreen trees or C4 grasses)". Please check section 2.2.**

*L235: Given these are deterministic model outputs, why conduct these statistic tests?*
*Only one instance of the atmospheric forcing, boundary conditions, parameters, and model structure is sampled, so what does it tell you if the difference between one model run or another is 'significant'? This might make sense if applied to ensembles of runs, but to compare one run against another it seems inappropriate.*

**Response: There are several sources of variability, such as the interannual climate variability (in the CV case) or spatial variability in the soils. Please, check the section 2.4 and 2.5**

*L275: This is over-stating the conclusions of the model. No real evidence is presented here that it correctly simulates the complex biophysics of forest flammability, so to draw this conclusion (that the model 'shows that the Amazonia(n) forest is naturally inflammable) is not defensible.*

**Response: Fair comment. We have modified this sentence in the manuscript.**

*L279: It seems strange to me that, in the absence of a detailed illustration that the model functions appropriately in these regions, there is no investigation of any type of within-model variability, and the structure and parameterizations of the model are assumed to be fixed. I see that this study aims to look at large modifications in model scope, but I find it unusual that no other model features are brought into question at all, particularly with regard to the strength of the conclusions.*

**Response: The modified manuscript addresses these issues more deeply.**

*L309: My reading of figure 5 is that the full model, with all elements, under-predicts biomass significantly over much of the transition region (transect 1 and 4 in particular).*
*Table 7 only presents correlations and not biases, so this feature is glossed over.*

**Response: We agree with the reviewer that there is an underestimate in biomass over much of the transition region. The purpose of the correlation analyses is to evaluate whether simulated and observed data are varying in the same direction and relative magnitude.**

*L313: How does this finding relate to those of the Senna and Castanho studies? This is too vague a reference.*

**Response: To clarify this, we have modified this sentence in the manuscript. Please check section 3.1.3.**

*L320: This is a highly complex system and biases can and do arise from a huge number of sources. It is not necessarily a local problem, nor anything to do with moisture stress those are both unfounded speculations.*
**Response: We agree with the reviewer. The sentence was removed from the manuscript.**

*L337: These conclusions - that phosphorus limitation and fire tend to reduce vegetation biomass, are pretty self-evident and not very interesting.*

**Response: The conclusions go beyond this point.**

*L339: The word 'robust' here is problematic. The model does not show deviate through time in these fields, but 'robust' is normally used to describe a simulation which is physically plausible, and I don't think that applies here necessarily.*

**Response: We disagree with the reviewer. A robust simulation, in the Computer Science literature, has been used to designate a simulation where small deviations (in this case by climate variability or by random events such as fire) does not cause the system to deviate from its expected behavior. If the result of the vegetation type simulation does not repeat itself in the last 10 years of the run, we consider that the simulation is not robust, and the vegetation is considered to be transitional. Same concept has been used by Pereira et al. (2013).**

*L363: Again, changing the model drastically 'led to significantly different average biomass' is not a very interesting conclusion from a piece of science. I do not think there is any debate about whether fire reduces forest biomass where it occurs, nor whether introducing a universally lower Vcmax might reduce vegetation productivity?*

**Response: We don´t understand what the reviewer mean by "changing the model drastically". We did not do such thing.**

*L369: Is climatic inter-annual variability in the CRU dataset realistic? There are other climate reanalysis that one might test it against, as well as station-level meteorological records. Given the incompleteness of met station data across this domain, it would seem likely that it underestimates variability somewhat.*

**Response: The reviewer may be right about this point, although one cannot prove it. Our previous experience with this dataset is that it is reasonable enough, considering the number of variables, resolution and time span. Other datasets may be better, but they may be specific or the time series is too short. If the CRU inter-annual climate variability is underestimated, it only means that the effects are greater than what we estimated.**
**The CRU database has been widely used by the scientific community in case studies, in validation of models due to their representativeness of the area (Li et al., 2016). There are no studies that invalidate the veracity of the interannual climate variability from the CRU. However, recent work by Beguería et al. (2016) suggests that spatial interpolation techniques used for constructing data are good at preserving only the mean of the data, and that they do not provide adequate representation of their variance, and this fact may lead to erroneous conclusions about changes in climate variability and extremes. We include this information in revised manuscript, please check Section 2.4.**

*L372: This is a very old reference for this very active field.*
**Response: We have included new references.**

*L413: How is the adaptation of savanna species to P-limitation represented in the model? As far as I can tell, the impact of P on Vcmax was universal and not PFT specific?*

**Response: We based our study on the fact that there is evidence that in Cerrado as well as in the Amazon, tree abundance has been generally associated with increases of soil fertility (Vourtilis et al., 2013), even as biodiversity richness and diversity (Long et al., 2012). Thus, we implemented the phosphorus limitation only in arboreal PFTs (tropical broadleaf evergreen tree and tropical broadleaf drought-deciduous tree).**
**To clarify this, we included information in revised manuscript, please check Section 2.2.**

*L426: These outputs do not actually show that understanding phosphorus limitation is associated with reliability of databases, it just shows that the databases are different.*

*An alternative model structure that is not so sensitive to overall soil P, for example, might conclude that the database discrepancy doesn't matter? That is a hypothetical example, but the logic of this sentence is unconvincing.*

**Response: Our text was not written properly which led to an erroneous interpretation. We rewrote this paragraph to convey the information correctly. Please check section 4.**

*L432: References to the state of the art in nutrient cycle modeling should be included here.*

**Response: Included.**

*L444: This logic (it is clear that phosphorus has a significant effect on woody biomass) is also unconvincing. It simply shows that the (simplistic) model predicts this, not that it happens in real life.*

**Response: We agree. To clarify, we rewrote this sentence to show that INLAND model can simulate the phosphorus effect on biomass.**

*L447: Again, this simply shows that this fire model does not burn the intact forest, and this cannot be used to conclusively state anything about real intact forest.*
**Response: This is has been changed in the revised manuscript.**

*L459: Are the physiological differences between cerrado and other vegetation types depicted in the CTEM model? Again, the sparse model description does not allow this to be determined.*

**Response: We provide a more detailed model description now.**

*L491: Given that there is no indication of how the parameterization for rainforest vegetation came to be in the first instance, one cannot say whether the P limitation should necessarily be an improvement. LAI in biosphere models can be modified trivially by changing the leaf lifespan and/or specific leaf area or leaf allocation scheme. All of these features are variable in observational datasets, and so the initial LAI predictions can, I am pretty certain, be modified massively. Whether the model over or underpredicts LAI in the first instance is therefore a matter of parameter choice, and therefore whether the P limitation improves or degrades the model is also a feature of that choice.*

**Response: Most DGVMs tend to overestimate LAI in the Amazon, and is not uncommon to find simulated values in the range of 10-12 $m^2$ $m^2$ in the literature. Of course, changing leaf parameters will change LAI, but one should be aware of using unrealistic parameters. P limitation reduces LAI values in general, but there are other factors affecting the final results.**

*L519: You predict that the vegetation distribution is affected by these things, not observe.*

**Response: That was changed in the manuscript.**

*L540: This is an extraordinarily grandiose and unneeded claim. I'm pretty sure that, for example, Levine et al. (2016) might disagree.*

**Response: The focus of Levine et al., (2016) is to evaluate the distribution of above-ground biomass across the Amazon basin using two independent satellite products, and not to evaluate the influence of climate variability, fire occurrence and phosphorus limitation in the Amazon-Cerrado transitional vegetation dynamics and structure. Still, to clarify possible conflict of interest, we have rewritten the sentence in the manuscript. Please check section 5.**

*L555: Bringing up the need for greater constraint on the uncertain model parameters at this point seems a bit too-little too-late.*
**Response: This is has been changed. We have rewritten the text and included this observation along the discussion.**

*Figure 1) I don't see how the transects are delineated in this figure?*
**Response: The figure was modified to improve clarity.**

[Figure]

*Figure 2) Definition of 'new' is ambiguous in the legend. As is the use of the '-' sign to denote PR and PG. Unclear if it means 'minus' or not.*
**Response: To clarify this, we have rewritten the legend.**

**Figure 2. (a) Map of regional total phosphorus in the soil (PR), (b) Map of global total phosphorus in the soil (Yang et al., 2013) (PG).**

*Figure 3) What is figure b? It doesn't say in the legend.*
 **Response: To clarify this, we have rewritten the legend.**

**Figure 3. West-East patterns of AGB in the Amazonia-Cerrado transition for transects T1, T2, T3 and T4 (a), and T5 (b) analyzed.**

*Table 7: Why only correlation coefficients and not also biases?*
**Response: We use the correlation coefficient because our aim is to evaluate the INLAND model ability to represent tendency of the observed data, and capture the biomass variability along the transitional areas.**

Correspondence to: Emily Ane D. da Silva (emilyy.ane@gmail.com)

**Abstract**

[revised manuscript text omitted]

In this paper we use the dynamic vegetation model INLAND (Integrated Model of Land Surface Processes) to evaluate the influence of inter-annual climate variability, fire occurrence and P limitation in the Amazon-Cerrado transitional vegetation dynamics and structure. We assess how each element affects the net primary production (NPP), leaf area index (LAI) and aboveground biomass (AGB) and compare the model simulated AGB to observed AGB data. The results presented here are important to build models that

accurately represent the transition vegetation, and show the need to include the spatial variability of eco-physiological parameters in these areas.

**2    Materials and methods**

**2.1    Study Area**

The present study focuses on the Amazon-Cerrado transition (Figure 1). We use the official delimitation of the Brazilian biomes proposed by IBGE (2004), and define five transects along the transition border. Transects 1 to 4 are established considering approximately 330 km into the Amazon and 330 km into the Cerrado domain, while Transect 5 is 880 km long on the southern Amazon-Cerrado border. The transects are located as follows: Transect 1 (T1, 43°- 49°W; 5°- 7°S), Transect 2 (T2, 46°-51° W; 7°-9S), Transect 3 (T3, 48°-54° W; 9°-11° S), Transect 4 (T4, 49° - 55° W; 11°-13° S), and Transect 5 (T5, 53° - 61° W; 13°-15° S) (Figure 1).

**2.2    Description of the INLAND Surface Model**

The Integrated Model of Land Surface Processes (INLAND) is the land-surface component of the Brazilian Earth System Model (BESM). INLAND is based on the IBIS model (Integrated Biosphere Simulator, Foley et al., 1996; Kucharik et al., 2000), which considers changes in the composition and structure of vegetation in response to the environment and incorporates important aspects of biosphere-atmosphere interactions. The model simulates the exchanges of energy, water, carbon and momentum between soil-vegetation-atmosphere. These processes are organized in a hierarchical framework and operate at different time steps, ranging from 60 minutes to 1 year, coupling ecological, biophysical and physiological processes. The vegetation structure is represented by two layers: upper (arboreal PFTs) and lower (no

arboreal PFTs, shrubs and grasses) canopies, and the composition is represented by 12 plant functional types (PFTs) (e.g., tropical broadleaf evergreen trees or C4 grasses, among several others).The photosynthesis and respiration processes are simulated in a mechanistic manner using the Ball-Berry-Farquhar model (details in Foley et al., 1996). The vegetation phenology module simulates the processes such as budding and senescence based on empirically-based temperature thresholds for each PFT. The dynamic vegetation module computes the following variables yearly for each PFT: gross and net primary productivity (GPP and NPP), changes in AGB pools, simple mortality disturbance processes and resultant LAI, thus allowing vegetation type and cover to change with time. The partitioning of the NPP for each PFT resolves carbon in three AGB pools: leaves, stems and fine roots. The LAI of each PFT is obtained by simply dividing leaf carbon by specific leaf area, which in INLAND is considered fixed (one value) for each PFT.

INLAND has eight soil layers to simulate the diurnal and seasonal variations of heat and moisture. Each layer is described in terms of soil temperature, volumetric water content and ice content (Foley et al., 1996; Thompson and Pollard, 1995). Furthermore, all of these processes are influenced by soil texture and amount of organic matter within the soil profile.

Considering these aspects of vegetation dynamics and soil physical properties the model can simulate plant competition for light and water between trees, shrubs and grasses through shading and differences in water uptake (Foley et al., 1996). These PFTs can coexist within a grid cell and their annual LAI values indicate the dominant vegetation type within a grid cell. For example, the dominant vegetation type is a Tropical Evergreen Forest if the PFT tropical broadleaf evergreen tree has an annual mean upper canopy LAI ($LAI_{upper}$) above 2.5 $m^2$ $m^{-2}$. On the other hand, the dominant vegetation type is a Tropical Deciduous Forest if the tropical broadleaf drought-deciduous tree has an annual mean $LAI_{upper}$ above 2.5 $m^2$ $m^{-2}$. Where total

tree LAI ($LAI_{upper}$) is between 0.8 and 2.5 $m^2$ $m^{-2}$, dominant vegetation type is savanna, and $LAI_{upper}$ values smaller than 0.8 $m^2$ $m^{-2}$ characterize a grassland vegetation type.

We assume that the vegetation types Tropical Evergreen Forest and the Tropical Deciduous Forest in INLAND represent the Amazon rainforest, while Savanna and Grasslands represent the Cerrado. Savanna would be equivalent to the Cerrado physiognomies *Cerradão* and *Cerrado sensu strictu*, while Grasslands would be equivalent to the physiognomies *Campo sujo* and *Campo Limpo* (*sensu* Ribeiro and Walter, 2008).

The soil chemical properties are represented by the carbon, nitrogen and phosphorus. The carbon cycle is simulated through vegetation, litter and soil organic matter, where the biogeochemical module is similar to the CENTURY model (Parton et al., 1993; Verberne et al., 1990). The amount of C existing in the first meter of soil is divided into different compartments characterized by their residence time, which can vary in an interval of hours for microbial AGB and organic matter to several years for lignin. The model considers only the soil N transformations and carbon decomposition, but the N cycle is not fully simulated and N does not influence the vegetation productivity, i.e., there is a fixed C:N ratio. P is used only to limit the gross primary productivity. The total P available in the soil ($P_{total}$) is used to estimate the maximum capacity of carboxylation by the Rubisco enzyme ($V_{max}$) through a linear relationship.

$$V_{max} = 0.1013\,P_{total} + 30.037 \qquad\qquad (1)$$

where $V_{max}$ and $P_{total}$ are given in $\mu molCO_2$ $m^{-2}$ $s^{-1}$ and $mg\,kg^{-1}$, respectively. This equation has been developed by Castanho et al. (2013) based on data for tropical evergreen and deciduous trees, and is applied only to these two PFTs in the model.

INLAND also contains a fire module, from the Canadian Terrestrial Ecosystem Model CTEM (Arora and Boer, 2005). In this module, three aspects of the fire triangle are considered – the availability of fuel to

burn, the flammability of vegetation depending on environmental conditions, and the presence of an ignition source. The natural ignition probability is summed to arbitrary anthropogenic fire probability, and the burned area is modeled as an ellipse of dimensions determined by wind and fuel conditions (Arora and Boer, 2005).

**2.3    Observed data**

**2.3.1    Phosphorus databases**

We used two P databases to estimate $V_{max}$ (Equation 1): one regional (referred to as PR) and one global database referred to as PG). In addition, a control P map (PC) represents the unlimited nutrient availability case, equivalent to a $V_{max}$ of 65 $\mu molCO_2$ $m^{-2}$ $s^{-1}$, or 350 mg P $kg^{-1}$ soil, according to Equation 1.

The PR database was developed from total P in the soil for the Amazon basin published by Quesada et al. (2011) plus 54 additional available P samples (P extracted via Mehlich-1 extractor, $P_{mehlich-1}$) (Figure 2a). We used the $P_{-mehlich-1}$ and clay contents measured in a forest-savanna transition region in Brazil (Mato Grosso state) to estimate $P_{total}$ and expand the coverage area of the P data (Section S1). These 54 samples were gridded to a $1° \times 1°$ grid to be compatible with the spatial resolution used by INLAND, resulting in 12 additional pixels with observed total P content (Figure 2a). For pixels without observed $P_{total}$, the $P_{total}$ was assumed to be 350 mg P $kg^{-1}$ soil, similarly to the PC conditions.

A global dataset of $P_{total}$ (Figure 2b) was also used to estimate $V_{max}$. This global data set is part of a database containing six global maps of the different forms of P in the soil (Yang et al., 2013). The $P_{total}$ was estimated from lithologic maps, distribution of soil development stages, fraction of the remaining source material for different stages of weathering using chronosequence studies (29 studies), and P distribution in different forms for each soil type based on the analysis of Hedley fractionation (Yang and Post, 2011), which are part of a worldwide collection of soil profile data. The uncertainties and limitations associated with this

database are restricted to the Hedley fractionation data used, which are 17% for low weathered soils, 65% for intermediate soils and 68% for highly weathered soils (Yang et al., 2013).

**2.3.2 Above-Ground AGB (AGB) database**

The AGB database used was created by Nogueira et al. (2015) and considered undisturbed (pre-deforestation) vegetation existing in the Brazilian Amazonia. This database was compiled from a vegetation map at a scale of 1:250000 (IBGE, 1992) and AGB averages from 41 published studies that had conducted direct sampling in either forest (2317 plots) or non-forest or contact zones (1830 plots). We bi-linearly interpolated the AGB (dry weight) for each transect considering $1° \times 1°$ to ensure compatibility of the observed and simulated data.

Five longitudinal transects (Figure 1) were used separately to characterize AGB in the Amazon-Cerrado border (Figures 3a and 3b). In T1, T2, T3 and T4, the higher AGB values in the west and lower values in the east are consistent with the transition from a dense and woody vegetation (the Amazon forest) towards a sparse vegetation with lower AGB (the Cerrado). However, T1 shows a more gradual reduction of AGB along the west to east gradient, while in T2, T3 and T4 where the transition is more abrupt. In T5 no west-east gradient is present with high AGB heterogeneity and predominant low AGB across the transect (Figure 3b).

**2.4 Simulations**

The model was forced with the prescribed climate data based on the Climate Research Unit (CRU) database (Harris et al., 2014). Two climate boundary conditions were used: the first is referred to as the

monthly climatological average (CA) that represents the average climate for the period 1961-1990. The second climate boundary condition is the historical dataset, for the continuous period between 1948 and 2008 (CV). For both boundary conditions, the variables used are rainfall, solar radiation, wind velocity and maximum and minimum temperatures. The CRU database has been widely used by the scientific community in case studies, because these data preserve the spatial mean of the rainfall data, although, they do not provide adequate representation of their variance precipitation (Beguería et al., 2016). The dataset has a 1-degree spatial resolution and a monthly time resolution.

Soil texture data is based on the IGBP-DIS global soil (Global Soil Data Task 2000) (Hansen and Reed, 2000). The model simulations were run for the time period 1582-2008, a total of 427 years. In the CV group of runs, the model was spin-up by cycling the 1948-2008 climate data (61-year) seven times, totaling 427 years. In the CA group of runs, the annual mean climate data was cycled 427 times. In both cases, $CO_2$ varied from 278 to 380 ppmv, according to observations in the period, updated annually. In both cases, only the model results of the last 10 years were used to analyze the results.

The experiment design is a factorial combination of the climate scenarios (CA, monthly climatological average, 1961-1990; CV, monthly climate time series, 1948-2008), the nutrient limitation on $V_{max}$ (PC, no P limitation ($V_{max} = 65$ $\mu molCO_2$ m$^{-2}$ s$^{-1}$); PR, regional P limitation; PG, global P limitation) and the occurrence of fire (F) or not (Table 1). The 12 combinations in Table 1 allow the evaluation of individual and combined effects of climate, soil chemistry, and the incidence of fire on the variables: Net Primary Production (NPP), tree AGB, and LAI of the upper and lower canopies ($LAI_{upper}$, $LAI_{lower}$).

We consider that the subtraction between the simulations $(CV+PC) - (CA+PC) = (CV-CA)|_{PC}$ represents the isolated effect of inter-annual climate variability without P limitations. The same logic is

applied to isolate other factors such as fire and P in different climate scenarios. For example, the fire effect under average climate without P limitation case is calculated by the difference between CA+PC+F and CA+PC, so that (CA+PC+F)–(CA+PC) = F|$_{CA,PC}$. Similarly, the isolated effect of fire under a climate with inter-annual variability scenario without influence of P limitation is calculated by the difference between CV+PC+F and CV+PC, so that (CV+PC+F)–(CV+PC) = F|$_{CV, PC}$. The different combinations of climate scenarios with and without fire effects and with and without P limitations are described in Table 2.

**2.5 Statistical analysis and determination of the best model configuration**

The statistical analysis is divided in four parts. First, we present maps of the isolated effects for all simulated area calculated as the average of last ten years of simulated spatial patterns. The statistical significance of the isolated effects on NPP, LAI and AGB are determined using the t-test with $p < 0.05$. The results are tested in each pixel, for all the simulated domain (n = 10).

Second, we present an analysis of variance using the one-way ANOVA and the Tukey-Kramer test in the transition zone. We consider all 31 pixels which fall in transects T1 to T5 ($n_{pixels}$). The results presented are based on the set of last 10 years of simulation (1999-2008, $n_{years}$) for the 12 combinations ($n_{simulation}$) in Table 1. Moreover, we grouped treatments according to climate regardless of P limitation, presence or absence of fire, where all sets with CV vs CA are tested (Group 1, n=1860, ($n_{pixel}$ x $n_{year}$ x ($n_{simulation}$/2))). Similarly, in Group 2 we tested if PC, PR or PG were significantly different from each other regardless the F or climate used (Group 2, n=1240, ($n_{pixel}$ x $n_{year}$ x ($n_{simulation}$/3))). In Group 3 we tested if fire introduced a significant effect regardless of climate and P limitation (Group 3, n=1860, ($n_{pixel}$ x $n_{year}$ x ($n_{simulation}$/2))). Finally, all treatments were tested to each simulation assessing their individual effects on NPP, LAI and AGB ($n_{pixel}$ x $n_{year}$ = 310).

Third, a correlation coefficient between the simulated and observed values for AGB was calculated for each transect. The simulated variables are averaged for the last 10 years of simulations (1999 - 2008) and compared to AGB from Nogueira et al. (2015) within a grid cell.

Finally, we evaluate INLAND's ability to assign the dominant vegetation type by analyzing 10 years of probability of occurrence. If the dominant vegetation type (evergreen tropical forest, or deciduous forest for the Amazon rainforest, and savanna or grasslands for Cerrado) in a pixel is the same in more than 90% of the simulated years (9 out of 10), then the simulated vegetation type is defined as "very robust" for that pixel; if it occurs in 70 – 90% of the simulated years, the simulated result is considered to be "robust". If the dominant vegetation occurred in less than 70% of simulated years, the pixel is considered "transitional" vegetation.

**3    Results**

**3.1    Influence of climate, fire and phosphorus in the Amazon-Cerrado transition region**

**3.1.1    Spatial patterns**

Overall, the inclusion of inter-annual climate variability (CV) resulted in a decrease in the simulated average tree biomass (TB) by 3.8% in Amazonia, and by 8.7% in Cerrado in comparison to average climate (CA) (Figure 4a). The spatial differences between CV and CA for TB simulations are statistically significant and range from -3 kg-C m$^{-2}$ to +2 kg-C m$^{-2}$. The state of Pará, with higher influence of the El Niño phenomenon, experienced the highest decrease in TB in the CV simulation. In the state of Roraima, on the other hand, there was an increase of about 2 kg-C m$^{-2}$ in TB when CV was considered. Bolivia and southwest

of Mato Grosso state also presented, in some grids points, a significant increase in AGB higher than 2 kg-C m$^{-2}$.

On average, P acts as a limiting factor in the simulated TB, decreasing by 13% in regional P (PR) simulation and 15% in global P (PG) simulation. In PR, TB decreased mainly in the southeastern Amazonia (between Pará and northeastern Mato Grosso states) and northwestern Amazonas state (Figure 4b). In PG, the largest TB decline occurred in central Amazonia, northeastern Pará and northeastern Mato Grosso (Figure 4c). In Cerrado, on the other hand, TB declined by 2% for PR and 9% for PG with respect to the control simulation. In PR, the few pixels in the Cerrado that have P limitation showed a significant decrease in TB (Figure 4b), while in PG the TB reduction was statistically significant for most of the Cerrado domain, except in southern Tocantins state (Figure 4c).

The tree biomass reduction due to fire events is much higher in magnitude more than due to P limitation or inter-annual climate variability (Figure 4d). The greater water availability is related to small or null fire effect in the Central Amazon rainforest agrees with the fact that Amazonia is naturally inflammable as well as a gradient towards seasonally dryer climate that increases the intensity and magnitude of fire effects towards the Cerrado (Figure 4d). The fire effect on TB over the Amazon domain was 21-24% of the P limitation effect (range for PR and PG cases), while the fire effect on TB over the Cerrado was more than 250% of the P limitation effects in CV simulations, which is due to quick growth of grasses after fire occurrence in the latter.

**3.1.2    Influence of climate, fire and phosphorus in the transects**

Results of the ANOVAs and Tukey-Kramer test indicate that the inclusion of CV, limitation by P (PR and PG) and fire in INLAND led to significantly different averages of NPP, LAI and AGB in the transition

zones. This influence of climate, P and fire are shown separately in Tables 3 to Table 5 and combined in Table 6.

The effects of climate and P on productivity show that CV reduces the NPP from 0.68 kg-C m$^{-2}$ yr$^{-1}$ to 0.64 kg-C m$^{-2}$ yr$^{-1}$ (Table 3) and the P effect results in NPP decline from 0.71 kg-C m$^{-2}$ yr$^{-1}$ to 0.64 kg-C m$^{-2}$ yr$^{-1}$ (both PR and PG) (Table 4). The fire effect, moreover, has a positive effect on NPP from 0.66 kg-C m$^{-2}$ yr$^{-1}$ when fire is off to 0.67 kg-C m$^{-2}$ yr$^{-1}$ when fires is on. This difference, albeit low, is statistically significant (Table 5).

In addition CV and P limitation reduce the LAI$_{total}$ in the canopy (Table 3 and Table 4), increasing three times LAI$_{lower}$ and decreasing LAI$_{upper}$ (Table 5). The magnitude of fire effect on AGB (46.7%, Table 5) is greater in relation to the CV (5%, Table 3) and P (14%, Table 4) limitation effects.

Even though CV effects on NPP and AGB for each simulation is not statistically significant, the effects of fire and P limitation (regardless of phosphorus map) are. Fire effects are significant only for structural variables as AGB, LAI$_{total}$, LAI$_{upper}$ and LAI$_{lower}$. It presents an increase of LAI total of 1.52 m$^2$ m$^{-2}$ in CV+PG+F in relation to CV+PG, and of 1.32 m$^2$ m$^{-2}$ in CV+PR+F in relation to CV+PR (Table 6).

**3.1.3   West-East patterns of AGB in the Amazon-Cerrado transition**

The model used in this study simulates > 80% of the observed AGB variability in all treatments along the transition area except in T5 (Table 7). It shows that the model is able to capture AGB variability along the transition area, which is relevant when compared to studies that simulate 50% of the observed AGB variability (Senna et al., 2009; Castanho et al., 2013).

It is not possible to identify a treatment that best represents AGB in all transects (Table 7). A combined analysis of Table 7 and Figure 5 indicates a general agreement that observed AGB decreases from W to E in

T1 to T4, and this is well captured by several configurations of the model, with specific differences among them. Overall, CA and PC configurations, being the least disturbed treatments, yield higher AGB, while the introduction of CV, PG and F reduce the AGB. However, the simulated results may be above or below the observed ones, which suggests that additional local factors are not included in the model.

The curves of AGB (Figure 5) show the impact of CV, PG and F along the W-E transition. PG has a high influence on the transition, decreasing the ABG especially in the western part of the transects, where the Amazon vegetation is predominant. This feature is particularly simulated in T3 and T4, where PG decrease the AGB by 2 kg-C m$^{-2}$ in the west pixels of these transects (Figure 5). In T1, T2 and T5, AGB decline is also higher with P limitation when compared to the curves limited only by CV. However, in T1 model simulations tend to underestimate the highest and the lowest AGB extremes, and the absolute values were always underestimated, despite the improvement in correlation with the inclusion of the fire component (Table 7).

Fire, however in T2, T3 and T4, is responsible to approach the simulated AGB to the observed AGB in the eastern pixels into Cerrado domain (Figure 5). In T5 these relations are similar, with climate presenting less influence on AGB decrease than P, and fire appears mainly as a reducer factor.

**3.2 Simulated composition of vegetation**

Most of the pixels in CA show very robust simulations, with more than 90% of the same vegetation cover in the simulated last 10 years (Figure 6a-c and 6g-i). A larger number of pixels with transitional vegetation were simulated in CV (Figure 6d-f and and 6j-l). An even higher variability in CV compared to CA simulations was observed when we added the effects of P limitation and fire (Figure 6a and 6j-l).

The vegetation composition in all P limitation scenarios for CA simulations resulted in robust simulations for nearly all pixels, except for the north of Cerrado domain (Figures 6a, 6b and 6c). The CA+PC and CA+PR simulations had the same vegetation composition, while CA+PG replaced the deciduous forest by evergreen forest in the central Cerrado region, around 8°S 46°W (Figures 6A, 6B and 6C). This behavior might be related to the higher $P_{total}$ values in PG than PR and PC for the Cerrado region (Figure S1). Cerrado was better represented in CV+PC, CV+PR and CV+PG than in the same CA combinations (Figure 6). The occurrence of forested areas in central Cerrado decreased in CV combinations, these being replaced by the savanna or grassland vegetation class.

When the effect of fire was added to CA simulations, the model simulated an increase in the uncertainty on the vegetation cover classification in the Cerrado region. The effect of fire reduced the presence of deciduous forest in central Cerrado biome as well as in CA+PC, and the vegetation was replaced by evergreen forest and savanna in CA+PC+F (Figures 6G, 6H, 6I). In CV simulations, fire effect results in the replacement of the deciduous and perennial forest by savanna and grasses in all central Cerrado region (Figures 6J, 6K and 6L).

For all combinations used, transitional forest areas in the northern and southwestern Cerrado biome are not adequately represented. With >90% of concordance, INLAND assigns the existence of tropical evergreen forest rather than deciduous forest in some pixels in the north of the transition, and the existence of tropical evergreen forest rather than savanna in the southwest, indicating difficulty to simulate transitional vegetation in these regions.

**4    Discussion**

The inclusion of CV, PR and PG and fire in INLAND showed significant influences on the simulated vegetation structure and dynamics in the Amazon-Cerrado border (Figure 4 and Table 6), suggesting that these factors play key role on vegetation structure in the forest-savanna border and can improve the simulated representation of the current contact zone between these biomes. This is broadly consistent with the literature that investigated causes of savanna existence in the real world (Hoffmann et al., 2012; Dantas et al., 2013; Lehmann et al., 2014). In this study, the spatial analysis and the Tukey-Kramer test (TK) show a difference in magnitude among these factors in vegetation, with fire occurrence and P limitation being stronger than inter-annual climate variability along the transects (Figure 4).

The spatial analysis showed that CV declines AGB predominantly in eastern Amazonia (Figure 4a). Climate of this region is intensely affected by El Niño–Southern Oscillation (ENSO), which could reduce precipitation by 50%, placing the vegetation under intense water stress (Botta and Foley, 2002; Foley et al. 2002; Marengo et al., 2004; Andreoli et al., 2013; Hilker et al., 2014). This reduction in rainfall in dry years brings in direct changes in carbon flux (NPP) and stocks in leaves and wood, leading to changes in vegetation structure. In addition to inter-annual changes in the rainfall, inter-annual variability in other climate variables in CV also affect AGB, as average, maximum and minimum temperature, as well as wind speed and specific humidity, and influence photosynthesis on the model both directly (through Collatz and Farquhar equations) and indirectly (e.g. through evapotranspiration). Our results showed significant differences for most part of the biomes, except central Amazonia (Figure 4a), where CV and precipitation seasonality have been pointed as secondary effects on vegetation (Restrepo-Coupe et al., 2013), since there is no shortage of water availability during the dry season.

Along the Cerrado, lower water availability in some years in CV affects tree biomass, although that vegetation is predominantly grassy-herbaceous. The AGB decline is significant for most part of the simulated Cerrado domain (Figure 4a) and average values could represent half the amount of typical tree biomass in this biome. This reduction in AGB reflects INLAND's ability to simulate similar Cerrado conditions and expose the few trees to high water stress.

Throughout the transects, however, no significant difference was found for average AGB between CV+PC and CA+PC by TK at $p<0.05$ (Table 6). On the other hand, when we analyzed the influence of CV for the same pixels, but using all simulations (Table 3), regardless of P limitation and fire occurrences, the results showed that the decrease in AGB by 0.38 kg-C m$^{-2}$ (5.7%) is statistically significant along the transition.

P limitation effect was statistically significant for PR and PG along all the Amazon domain and the main differences between these simulations were the spatial patterns of tree AGB decrease (Figure 4b and Figure 4c). We cannot affirm which of these databases is better because they are the results of different methodologies and observations (Quesada et al., 2009; Yang et al., 2014). However, PG showed a higher AGB decrease in central Amazonia, northeastern Pará and northeastern Mato Grosso state, indicating that in these areas the P limitation is higher. This result does not corroborate the northwest-southeast AGB gradient found in the Amazon basin, which showed a higher productivity in the west where soils are more fertile than those found in the southeast (Aragão et al., 2009; Saatchi et al., 2007; Nunes et al., 2012; Lee et al., 2013). On the other hand, PR AGB agrees with the northwest-southeast gradient, presenting less limitation in the soils of central Amazonia with declines in AGB mainly in the southeastern part of the rainforest (between Pará and northeastern Mato Grosso states) (Figure 4b).

In Cerrado, P limitation also influenced vegetation (Figure 4c) and presented statistically significant differences when compared to CV+PC. In this biome, as well as in the Amazon, tree abundance richness and diversity have been generally associated with increases of soil fertility (Long et al., 2012; Vourtilis et al., 2013), highlighting the importance of P in the composition and maintenance of vegetation, especially in transition areas.

Compared to the Amazon domain, the magnitude of effects of P limitation is lower in the Cerrado. However, few pixels in PR that have P limitation showed a significant decrease in arboreal AGB (Figure 4b), while in PG, we found reduction of AGB for most of the Cerrado domain, except only for the southern Tocantins state (Figure 4c). Despite the differences in spatial patterns, there was no statistically significant differences between PR and PG within the transects (Table 4 and Table 6).

The spatial difference between PG and PR showed that PG is lower than PR in the western Amazonia, and higher in northern Amazonia. Moreover, PG have low P values in south of the transition compared to PR, while in Cerrado domain P values ranged between 120 to 200 mg kg$^{-1}$ (Figure S1). Although the PR dataset includes every known P data collected in the region, these differences reinforce the need to improve the data of $P_{total}$ in the soils of the Amazon and Cerrado/Amazon transition domains. Currently, $P_{total}$ data in Cerrado is scarce, and make unfeasible to establish a proxy similar to Castanho et al. (2013), which was specific for the Amazon.

To our knowledge, the most part of the Dynamic Global Vegetation Models (DGVMs) do not consider the complete phosphorus cycle, despite the importance of nutrient cycling for AGB maintenance and tropical vegetation dynamics in dystrophic soils. For example, nutrient cycling in the Amazon/Cerrado transition is closely related to the hyper-dynamic turnover of the AGB (Valadão et al. 2016), in which some key species

might also be crucial to the hyper-cycling of nutrients through which vegetation sustain the constant input of nutrients, including large annual amounts of available P (Oliveira et al. *in press*).

The decrease in tree AGB occasioned by P limitation can contribute to a decrease in litter production and consequently could affect nutrient cycling in tropical ecosystems. According to Oliveira et al. (*in press*), the litter produced by vegetation corresponds to the main return route to the available fraction of P for plants, especially in the transition areas, where $P_{available}$ in the soil is very low. In our model, however, P acts directly in the photosynthesis limitation through $V_{max}$ and cannot be reabsorbed by the roots. Thus, the litter produced in vegetation contribute only to dry matter and fire occurrence increase. In nature, the litter affected by fire occurrence volatilizes the small amount of P available to plants, increasing the nutrient losses of the ecosystem. Despite this simplified representation in INLAND, it can represent the P influence on woody AGB in the Amazon and Cerrado.

The fire occurrence is an important factor controlling the AGB dynamics in the Cerrado or in the transition vegetation (Silvério et al., 2013; Couto-Santos et al., 2014; Balch et al., 2015), which this study clearly replicates, showing statistically significant influences when compared to control simulations (Figure 4d and Table 5). In the transition, the fire effect may reduce average AGB by 50%, which under climate change or deforestation conditions may lead to an even stronger change in the vegetation structure and dynamic. In the Cerrado domain, the simulated fire effect implies in significant increases of shrubs and herbaceous vegetation and decreases of the arboreal component. In nature, however, the Cerrado is relatively resilient to fire depending on the velocity, intensity and duration of the burning (Rezende et al., 2005; Elias et al., 2013 Reis et al., 2015). The adaptive morphological nature and the low nutrient requirement of vegetation allow Cerrado the capacity to rapidly restore the vegetation after fire occurrence (Hoffmann et al.,

2005; Hoffmann et al., 2012). In our model, the restoration of vegetation after fire occurrence is exclusively due to the canopy opening and consequently more luminosity penetration into lower canopy.

This study shows an improvement in the correlations between simulated and observed AGB when compared to previous modeling studies, regardless of treatment, with correlation coefficients usually above 0.80 for the transects, except for T5, for which the correlation coefficient value is usually below 0.5 (Table 7). Senna et al. (2009) found 0.20 as maximum correlation coefficient between simulated and observed ABG while Castanho et al. (2013) showed 0.80 for Amazonia domain. From Figure 5, it is clear that CV, F and P limitation in the transition zone reduce the AGB, approaching the simulated to the observed data, and play important roles in the simulations, but the only inclusion of these effects is still insufficient to represent the actual vegetation structure in the Amazon-Cerrado border (Figure 6L). In our interpretation, this means that other important factors are still missing from the simulation, especially in T5, where soils are rocky and shallow. A better spatial representation of soil physical properties, including shallow rocky soils, as well as spatially varying physiological parameterizations of the vegetation such as carbon allocation, deciduousness of vegetation, and residence time are probably needed to improve the simulations, in particular in the northern and southern extremes of the border (T1 and T5).

In addition, literature shows that in the transition area, soils are very different than Amazon soils, and that essential proprieties for modeling are peculiar (Silva et al., 2006; Vourlitis et al., 2013; Dias et al. 2015). For example, Dias et al. (2015) recently showed that the pedological functions normally used by DGVMs may underestimate the saturated hydraulic conductivity ($K_s$) by >99%, transforming a well-drained soil with $K_s = 1.5.10^{-4}$ m.s$^{-1}$ (540 mm.h$^{-1}$) in reality into an impervious brick with $K_s = 3.3.10^{-7}$ m.s$^{-1}$ (1.2 mm.h$^{-1}$) in the model.

For all transects, the AGB curves have similar patterns (Figure 5); the smaller difference is observed between CA+PC and CV+PG curves, while the larger difference is when fire is present. The effect of P limitation appears as an effect of intermediate magnitude, reducing the AGB by more than the effect of inter-annual climate variability. In the east, it is observed that there is little or no difference among AGB simulated by CA+PC, CV+PC and CV+PG, revealing that inter-annual climate variability and P have smaller influence in the AGB. However, in the east of T2, T3 and T4, fire is the factor that adjusts the simulated to the observed data (Figure 5), differently than the grid points in the West, where CV+PG is a better proxy between observed and simulated data.

Such conditions are interesting because they reflect the different mechanisms that regulate the structure of these ecosystems and probably the phytophysiognomies distribution. For example, P limitation seems to be the factor that improves simulated AGB in regions where the predominant vegetation type is the tropical rainforest. Fire, on the other hand, improves the AGB in grid points where the Cerrado occurs. Moreover, important factors such as productivity partitioning into leaves, roots and wood carbon pools are assumed to be fixed in space and time within a given PFT, neglecting the natural capacity of transitional forests to adapt itself and to adjust their metabolism to local environmental conditions (Senna et al., 2009). In years of severe drought, transitional forests could prioritize the stock of carbon to fine roots instead of the basal increment to maximize access to available water, or make hydraulic redistribution to maintain the greenness and photosynthesis rates. Brando et al. (2008) found high sensitivity in carbon allocation for eastern Amazon basin trees, which reduced wood production by 13-60% in response to an artificial drought. Although in INLAND soil moisture can reduce the photosynthetic rates during the months of lower rainfall,

it does not dynamically change the allocation rates, exposing the PFTs in these areas to severe water stress and underestimating the AGB, such as in the west of T1 (Figure 5a).

T2, T3 and T4, located in the central part of the Amazon-Cerrado transition, showed the highest average correlations between observed and simulated data (Table 7). For these transects, INLAND seems to be able to capture the high variability of AGB gradient.

At T5, located at the south of the transition, the average correlations were low for all treatments, indicating that INLAND has difficulty to represent the AGB gradient there (Table 7). However, it captures the lower AGB as compared to the northern ones. In this region, the vegetation is characterized by a wide diversity of physiognomies, which varies with other preponderant factors, such as lithology, soil depth, topography and fertility. The observed data also showed high AGB variability, indicating that there are changes in the vegetation structure, featuring medium-sized and small vegetation types on different soil types. In INLAND, however, features such as lithology and water-table depth are not considered due to the complexity of its representation on the large scale, limiting the representation of a heterogeneous environment throughout the transition.

Different patterns of vegetation distribution along the Amazon-Cerrado border exist and are influenced not only by inter-annual climate variability, P limitation, and fire, but also by the ecophysiological parameters, which may have different behavior according to the environmental conditions and soil proprieties. Obtaining these parameters is a challenge to the scientific community once the field measurements are difficult due to the extension of the transition area. More observed data are needed to establish and implement the plasticity of the fixed parameters such as carbon coefficients allocation, residence time, dependence of the deciduousness on P, among others.

Another point to discuss is that the model simulates, in a few pixels in southeastern Cerrado, very robust simulations of the presence of savanna and grassland even in the absence of fire (Figure 6A-F and 6a-f). This is, in our view, a result of the intense water and heat stress in this region. In the Brazilian Cerrado, the high temperatures (> 35 °C) combined to the dry season duration (as long as 6 months with little or no rain) exposes the vegetation to a severely stressed situation, so that a low biomass, low LAI vegetation may exist without the need of a frequent disturbance.

**5   Conclusions**

This is the first study that uses modeling to assess the influence of inter-annual climate variability, fire occurrence and phosphorus limitation to represent the Amazon-Cerrado border. This study shows that, although the model forced by a climatological database is able to simulate basic characteristics of the Amazon-Cerrado transition, the addition of factors such as inter-annual climate variability, phosphorus limitation and fire gradually improves simulated vegetation types. These effects are not homogeneous along the latitudinal/longitudinal gradient, which makes the adequate simulation of biomass challenging in some places along the transition. Our work shows that fire is in the main determinant factor of the vegetation changes along the transition. The nutrient limitation is second in magnitude, stronger than the effect of inter-annual climate variability.

[revised manuscript text omitted]